# Glacial limitation of tropical mountain height

**Maxwell T. Cunningham[1,2]\*, Colin P. Stark[2], Michael R. Kaplan[2], Joerg M. Schaefer[1,2]**

*[1]Dept of Earth and Environmental Sciences, Columbia University, New York, New York, 10027*

*[2]Lamont-Doherty Earth Observatory of Columbia University, Palisades, New York, 10964*

*Correspondence to: Maxwell T. Cunningham (maxwellc@ldeo.columbia.edu)*

**ABSTRACT**

Absent glacial erosion, mountain range height is limited by the rate of bedrock river incision, and is thought to asymptote to a steady-state elevation as erosion and rock uplift rates converge. For glaciated mountains,

there is evidence that range height is limited by glacial erosion rates, which vary cyclically with glaciations. The strongest evidence for glacial limitation is at mid-latitudes, where range-scale hypsometric maxima (modal elevations) lie within the bounds of late-Pleistocene snowline variation. In the tropics, where mountain glaciation is sparse, range elevation is generally considered to be fluvially limited and glacial limitation is discounted. Here we present topographic evidence to the contrary. By applying both old and

new methods of hypsometric analysis to high mountains in the tropics, we show that (a) the majority are subject to glacial erosion linked to a perched base-level set by the snowline or equilibrium line altitude (ELA), and (b) many truncate through glacial erosion towards the cold-phase ELA. Evaluation of the hypsometric analyses at two field sites where glacial limitation is seemingly marginal reveals how glacio-fluvial processes act in tandem to accelerate erosion near the cold-phase ELA during warm phases and to reduce their

preservation potential. We conclude that glacial erosion truncates high tropical mountains on a cyclic basis: zones of glacial erosion expand during cold periods, and contract during warm periods as fluvially-driven escarpments encroach and destroy evidence of glacial action. The inherent disequilibrium of this glacio-fluvial limitation complicates the concept of time-averaged erosional steady-state, making it meaningful only on long time-scales far exceeding the interval between major glaciations.

# 1 Introduction

The height of non-glaciated mountain ranges is limited by erosion along rivers, with rock uplift steepening channels and accelerating channel incision until, in principle, both rates match. This mechanism ties mountain erosion to a low-elevation base-level, and channel incision abruptly switches to deposition at this elevation. However, in many mountains there is a transition at high elevations from fluvial to glacial conditions, which introduces a second base level perched close to the glacial equilibrium line altitude (ELA). Above the ELA, rates of glacier incision and coupled supra-ice rock-slope erosion may match or exceed rock uplift rates; below the ELA, in the ablation zone, sub-glacial incision rates asymptote to zero while fluvial erosion is suppressed. Under these conditions, erosion towards the perched glacial base-level is the essential height-limiting mechanism.

For well over a century (Penck, 1905) it has been thought that such glacial limitation of mountain height is widespread. In the late 20th century, the concept of glacial limitation was taken to an extreme, and reframed as the "glacial buzzsaw" hypothesis (Brozović et al. 1996). The hypothesis is rooted in the observation that mountain ranges high enough for Pleistocene glaciation do not usually rise much higher than the ELA (Fig. 1), and it claims that glacial erosion imposes a near-global topographic ceiling by cutting large swaths of terrain down to this elevation. It has been an important contribution to the broader realization that landscapes evolve under the interacting influence of tectonically-driven crustal deformation and climatically-modulated erosion (e.g., Molnar and England, 1990; Willett et al., 2006). Debate over the prevalence (or even existence) of a glacial buzzsaw cuts to the core challenge of disentangling climatic and tectonic imprints on landscapes.

The strongest proponents of a global glacial buzzsaw have implied that erosion can outpace strong tectonic forcing once uplifted rock mass reaches the glacial equilibrium line altitude (ELA)—a climatically-determined elevation. Detractors claim that the power of the glacial buzzsaw has been overstated, and instead suggest that in most cases glaciations have ornamented, rather than substantially cut, uplifted terrain, which remains at high elevations precisely because erosion is unable to remove it (van der Beek et al., 2009; Hall and Kleman, 2014).

Here, we present a new test of the hypothesis that the height of mountains with terrain above the ELA is set by a perched *de facto* glacial base-level. We focus on a group of tropical mountains whose erosion is dominated by fluvial processes driven by and tightly linked to tectonic uplift, that are unaffected by recent volcanic construction and free of broad areas of internal drainage, and that only encounter the ELA at high elevations. Our rationale is that these mountains will be particularly sensitive recorders of glacial limitation

precisely because non-glacial erosion processes have been imprinted on them so strongly. In other words, we assess the potential for glacial limitation in places where fluvial limitation is, *a priori*, more likely. We ultimately find evidence in nine ranges that glacial erosion limits mountain height. In the most marginally glaciated ranges, such as the Central Range of Taiwan and the Talamanca Range of Costa Rica, glacial and fluvial erosion work in tandem to limit range height to the ELA – a process we term "glacio-fluvial limitation". In contrast, the perched glacial base-level is fully developed in those ranges where the most rock mass has been advected through the ELA. This surprising result adds new context to the concept of glacial limitation by showing that a high elevation, glacial base-level has been periodically introduced even in the warmest in places.

## 1.1 What does *glacial limitation of mountain height* mean?

Glacial limitation refers to the presence of an erosional base-level at the ELA and the disconnection of glaciated terrain from fluvial base-level forcing (Egholm et al., 2009). The introduction of this perched base-level results in the widening of glacial terrain near the ELA, and does not necessarily affect the distribution of peak heights, which are often found on isolated spires far above the ELA (e.g., Brozović et al., 1997; Ward et al., 2012). The concept of base level is usually associated with fluvial erosion, in which the rate of river incision undergoes a zero-crossing, typically at an elevation where channelized flow becomes unconfined and spreads laterally, such as at sea/lake level or at a piedmont-alluvial fan apex (Fig. 2a). Above fluvial base-level, river channel incision drives hillslope erosion and competes with rock uplift to set range relief. Erosional behavior above and below the perched, glacial base-level is analogous. Below the ELA, ice flow spreads laterally, ablates, and slows, driving sub-glacial erosion rates to zero. The (near-) ELA acts as an erosional base-level, above which ice-driven erosion pushes headward into the landscape (Fig. 2b). Glacial erosion ultimately disconnects these landscapes from fluvial base-level by blocking channel incision above the glacier terminus—an elevation where glacial erosion is also least effective. During glaciations, continued channel incision below the glacier terminus amplifies the disconnection between fluvial and glacial landscapes.

In mountains thought to be glacially limited, it is common to find peaks that rise more than 1 km above the ELA (Brozović et al., 1997). Glacial limitation allows such high peaks if they are coupled to cirque/valley glacier incision that is effectively controlled by the ELA. The concept is similar to fluvial limitation, such that high peaks rise several kilometers above fluvial base-level but are ultimately limited in height by the efficacy of erosion. In the glacially-limited case, glacial erosion introduced near the ELA lowers mountains that would otherwise reach higher elevations in fluvially-limited conditions.

## 1.2 Motivation: *Does glacial erosion limit tropical mountain height?*

Tropical mountains have traditionally been considered to be fluvially limited. Their high rates of rainfall drive high rates of fluvial incision and coupled hillslope erosion, and these processes are generally thought to dominate landscape evolution. Although such mountain ranges have been glaciated repeatedly throughout the Pleistocene (Porter et al., 1989; Hastenrath, 2009), the erosional consequences of snowfall and glacial ice have typically been deemed negligible and limited to very high elevations. Glacial erosion has therefore been discounted as a factor in limiting the height of tropical mountain ranges—with only a few exceptions (e.g., Ring, 2008; Margirier et al., 2016). Our goal in this paper is to assess the possibility of glacial limitation in tropical ranges, and by extension to reevaluate the efficacy of glacial limitation in general.

## 1.3 Structure of this paper

We explore the possibility that glacial erosion has limited the rise of tropical mountains by looking closely at the morphology of tropical highlands. First, we review the evidence for glacial limitation globally, and highlight how hypsometry has been the tool of choice in assessing glacial limitation. We then focus on the tropics and undertake topographic and geomorphic analysis of a broad selection of high tropical mountain ranges. We carry out range-scale hypsometric analysis of ten such ranges to evaluate signs of correlation between mountain height, the ELA, and glacial erosion. Next, we develop a new methodology for analyzing catchment hypsometry on a progressive, nested fashion, which helps with the comparison of hypsometry in mountain ranges of very different sizes. Armed with insights from these analyses, we evaluate them in detail in the Talamanca Range of Costa Rica and the Central Range of Taiwan. These two ranges were chosen because of the seemingly marginal influence of glacial erosion in limiting their height. We find evidence to the contrary: the data suggest that glacial erosion has played a major role in the long-term evolution of both landscapes, but that evidence of its role has been compromised by the headward encroachment of surrounding fluvial terrain during interglacials. The poor preservation potential of glacial morphology in the tropics has led to an underestimation of its importance in limiting range height.

## 2 Evidence for glacial limitation: a review

### 2.1 First reports

The first articulation of the idea that ice-driven erosion could limit mountain range height came over a century ago, when Dawson (1895) asserted that frost-cracking had kept the ridgeline of the Kamloops, British Columbia, at a relatively constant elevation. His claim was speculative and called attention to a striking similarity in the elevation of mountain summits, and their apparent correlation with the bottom of the frost-cracking window. Penck (1905) put forth a more formal argument for glacial limitation of mountain height

around the same time, but instead focused on sub-glacial erosion. In this remarkable paper, Penck described the differences between glacial and fluvial valleys in the Western Alps, and proposed that glacial erosion had flattened and widened pre-existing fluvial valleys to accommodate ice flux. Penck reasoned that the topographic structure of the Western Alps is geologically young, and (in the absence of any plate tectonics framework) hypothesized that glacial erosion had outpaced the rate of rock uplift and limited the height of

the range. It was not until the late 20$^{th}$ century, however, that the potential for widespread limitation of mountain height by glacial erosion was considered, when Porter (1977) and Broecker and Denton (1989) used a compilation of ELA estimates (paleo and modern) from the length of the American Cordillera to demonstrate a qualitative match between peak mountain height and the bounds of Pleistocene ELA fluctuation.

### 2.2 Emphasis on mid-latitudes

The modern form this idea is the glacial buzzsaw hypothesis, which was first articulated by Brozović et al. (1996, 1997). They found that hypsometric maxima (modal average elevations) of large (~1000 km$^2$)

physiographic provinces in the northwest Himalaya lie just below the modern snowline (used as a proxy for the modern ELA), despite major differences in erosion rate, geologic structure, and local climate. Brozović et al. (1997) made the important conceptual advance of defining mountain height as the hypsometric maximum of a large region, and showed that even though peak elevations of different physiographic provinces in the northwest Himalaya differ by more than 1 km, the hypsometric maxima of these provinces

are similar and match the altitude of the ELA. They attributed this topographic pattern to headward erosion focused at the ELA by cirque glaciers, a process that leaves low-gradient topography near the ELA, and, in many cases, large rock spires that rise above cirque floors.

In the years since, this hypothesis has gained widespread although incomplete acceptance. A hypsometric

maximum within the bounds of late-Pleistocene ELA variation is now widely regarded as the signature of

the glacial buzzsaw (e.g., Brozović et al., 1997; Brocklehurst and Whipple, 2004; Oskin and Burbank, 2005; Mitchell and Montgomery, 2006; Egholm, et al., 2009; Mitchell and Humphries, 2015), although *how* glacial erosion produces this hypsometric maximum is debated. There is broad agreement (Anderson et al., 2006; Egholm et al., 2009; MacGregor et al., 2009; Pedersen and Egholm, 2013) that cirque glacier erosion is ultimately responsible. Egholm et al. (2009) demonstrated that the small drainage area of cirque glaciers ultimately limits their ability to incise below the ELA, even if the large trunk glaciers they feed can erode far below this elevation (Valla et al., 2011). The ELA thus acts as an erosional base-level for tributary cirque glaciers. When the glacial base-level is effective over a large area, it is recorded in the landscape as a hypsometric maximum near the ELA.

Interpretation of the match between landscape hypsometric maximum and the ELA often links the hypsometric maximum to low-gradient terrain left near the ELA by cirque-driven headward erosion (e.g., Brozović et al., 1997; Egholm et al., 2009). It should be noted, however, that hypsometry is simply a measure of the fraction of the terrain covered by each contour band. A hypsometric maximum is not uniquely determined by topographic gradient: rather, the hypsometric maximum should be thought of as a contour whose combined length and thickness exceeds all other contours. A hypsometric maximum at the ELA is consistent with a lengthening of the ELA contour by headward cirque erosion as well the presence of low-gradient terrain at this elevation. In most cases, the hypsometric maxima found at the ELA probably arise both from contour width (low-gradient terrain) and contour length (erosional penetration) at the ELA.

The strongest empirical evidence for widespread glacial limitation comes from the global analysis of Egholm et al. (2009), who analyzed large swaths of topography (1°x1° tiles of Shuttle Radar Topography Mission digital elevation) to show that on the scale of a mountain range, hypsometric maxima almost never lie above the upper limit of late-Pleistocene ELA fluctuation. They showed that most mountain ranges affected by Pleistocene glaciation have a hypsometric maximum between the upper and lower bounds of ELA variation (Fig. 1), similar to the observations made by Brozović et al. (1997) in the northwest Himalaya. It is striking how widely this bound is observed over an ELA span of ~5 km.

Another line of evidence for glacial limitation has been an apparent acceleration in exhumation rate that coincides with the onset of Pleistocene glaciation. The thought is that if mountain ranges around the world have been subject to glacial limitation, a corresponding increase in exhumation rate should be apparent close to the onset of global glaciation in the thermochronometric record. This signal has been borne out on a regional level, such as in the Southern Andes (Thomson et al., 2010), Western Alps (Fox et al., 2014), and Cordillera Blanca (Margirier et al., 2016). Thus, the evidence for glacial limitation consists of both an

apparent glacially-driven increase in exhumation rate recorded in thermochronometry as well as a near-global topographic signature.

The idea that glacial erosion limits mountain height, and that mountains would generally be higher in an ice-free world, is not universally accepted. A key problem is that it is difficult to disentangle the topographic signatures of non-glacial versus glacial processes (e.g., Hall and Kleman, 2014). Ironically, one of the most problematic cases comes from the birthplace of the glacial buzzsaw hypothesis, in the northwest Himalaya, where Van der Beek et al. (2009) used a combination of thermochronometry and topographic analysis to propose that Pleistocene glaciation took advantage of a pre-existing eroded surface—and that the correlation between region-scale hypsometric maxima and the Pleistocene ELA is simply a coincidence of low-gradient topography near the ELA. It has been proposed that this kind of topographic inheritance may comprise the majority of the global signature of glacial limitation (Hall and Kleman, 2014). Another matter of significant controversy is whether mountain ranges were actually subject to enhanced erosion rates during the Pleistocene. Schildgen et al. (2018) analyzed 30 locations where enhanced Pleistocene exhumation rates had been inferred from thermochronometric data (Herman et al., 2013) and found that the signal of enhanced exhumation in 27 of those sites could be explained either by bias introduced by inappropriate sampling or by a change in tectonic boundary conditions, rather than by climate change. Inferences about global-scale exhumation patterns thus merit reevaluation. Our paper does not directly address these issues, and instead evaluates the topographic imprint of glacial erosion, focusing on locations where it has not been recognized before.

## 2.3 In the tropics

The principal reason for skepticism of glacial limitation in the tropics is that regional hypsometry shows: (a) very little area near the lower-limit of the Pleistocene ELA, and (b) significant hypsometric maxima several km below it (Fig. 1). Since most of these mountains are ridge-and-valley landscapes dominated by fluvial incision, it is reasonable to assume that if glacial erosion has taken place at all, it has only affected isolated portions of the mountain range. A similar rationale is that these ranges need to be well above 3000 m for cold-phase glacial erosion at all, and thus most glacial landscapes in the tropics sit above quickly eroding highlands whose height is thought to be controlled by the rate of fluvial incision. Nevertheless, the observation that mountain height rarely exceeds the ELA includes the tropics (Broecker and Denton, 1989; Egholm et al., 2009).

# 3  Study areas

### 3.1.1  Site selection

Seeking to broadly assess the potential for glacial limitation in tropical mountain ranges, we reviewed all those ranges close to the height of the cpELA. Within this broad selection, we sought a subset uncontaminated by processes that would distort or complicate any signal of glacial limitation. In particular, we wanted to avoid any confusion between the passive uplift of low-relief terrain from that of *in-situ* glacial erosion. The potential for such confusion is strongest in the Peruvian/Bolivian Andes, the Sierra Madre of Mexico and Guatemala, the central highlands of Papua New Guinea, all of which are characterized by large, internally drained plateaus disconnected from external base-level; as such, these regions were excluded from our analysis. Volcanism is another complicating factor, in that glaciated volcanoes have undergone a mix of construction and erosion that cannot easily be disentangled. Therefore, glaciated volcanoes such as those in East Africa (Mt. Kilimanjaro and Mt. Kenya) and Papua New Guinea (Mt. Giluwe) were excluded from our analysis. A final constraint was that each range must be circumferentially well-connected to external base-level (sea-level or lake-level) by relatively short fluvial links.

The following ten tropical mountain ranges remain and were selected for analysis (Fig. 3):
1) Leuser Range, Aceh, Indonesia
2) Central Range, Taiwan
3) Talamanca Range, Costa Rica
4) Crocker Range, Borneo
5) Finisterre Range, Papua New Guinea
6) Owen Stanley Range, Papua New Guinea
7) Merauke Range, Papua
8) Mérida Range, Venezuela
9) Sierra Nevada de Santa Marta, Colombia
10) Rwenzori, East Africa

### 3.1.2 Comparing the ELA across tropical mountains

For our tropics-wide comparison we used an ELA range of 3400-4000 m, which represents the approximate vertical span of the LGM ELA estimated around the tropics (e.g., Mark et al., 2005; Hastenrath, 2009), and is similar to the vertical range of LGM ELA in particularly wide mountain ranges with strong spatial variation in local climate (e.g., Merauke Range, Papua: Prentice et al., 2005; Mérida Range, Colombia: Stansell et al., 2007). We adopt the term cold-phase ELA (cpELA) to emphasize that the ELA has repeatedly descended to

roughly this elevation in tropical mountain ranges during glacial periods in the late-Pleistocene (e.g., Farber et al., 2005; Barrows et al., 2011), regardless of whether mountains were high enough to intersect it. Importantly, the cpELA is not strictly interchangeable with the gLGM ELA, since the timing of the local LGM appears to have varied substantially (10-20 kyr) in the tropics and subtropics (e.g., Ono et al., 2004).

Furthermore, ELA estimates are based on geomorphic reconstructions that ignore tectonics: as such they are potentially biased towards higher elevations by post-glacial rock uplift. Where glacial landforms are dated to the LGM such bias is likely negligible. When age constraints are lacking, and an LGM age for the glacial landscape is in doubt, ELA estimates may be influenced by pre-LGM glacial landforms subject to rock uplift for longer periods of time: in these cases, the risk of bias towards higher elevations (>100 m) is significant.

However, our goal is to assess whether mountains prone to glaciation have been limited by glacial erosion, regardless of when or at what elevation such erosion took place. As long as the cpELA has repeatedly descended to a similar elevation during the late-Pleistocene, error in its estimation on the scale of hundreds of meters is too small to compromise any assessment of whether mountain height has been limited at an elevation of several kilometers.

On a global basis, the ELA of interglacial periods (which for convenience we term the "warm-phase" ELA or wpELA) has generally been 800-1000 m above the cpELA (Porter, 1989; Broecker and Denton, 1989). We consider the wpELA to be broadly interchangeable with the modern ELA, although warming and commensurate glacier retreat during the late 20th century complicates the definition of the modern ELA. Four

of our ten selected mountain ranges are currently glaciated and thus intersect the wpELA: the Merauke Range, the Rwenzori, the Sierra Nevada de Santa Marta, and the Mérida Range. In the Merauke Range, the only remaining glacier is the Carstenz Glacier, where Allison and Kruss (1977) and Prentice et al. (2005) both estimated a modern ELA of 4650 m using aerial photographs (Prentice et al. (2005) found that ice loss has accelerated at the Carstenz Glacier since the 1970s, and suggested that the most reliable modern ELA estimate

was based on 1972 imagery). In the Rwenzori Mountains, Kaser and Osmaston (2002) compiled aerial photographs and observations from several field expeditions to map changes in glacier extent between 1955 and 1990. They used field mapping and aerial photographs from the 1950s and 1960s to estimate a modern ELA of 4600-4700 m. They noted that ice loss has also accelerated in the Rwenzori during the study period. Less work on wpELA estimation has been done in the Sierra Nevada de Santa Marta. Wood (1970) mapped

glacier extent there using aerial photographs acquired in 1969. He compared his results to the mapping efforts of the 1939 Cabot Expedition (Cabot et al., 1939) and demonstrated significant ice retreat during this time. No wpELA estimate was provided. Modern glaciers are presently less than 2 km$^2$ in the Mérida Range (Stansell et al., 2007) and have been steadily shrinking throughout the 20th century (Schubert et al., 1992).

Stansell et al. (2007) used the elevation of the 0° isotherm to broadly constrain the wpELA in the Mérida Range to 4470-5040 m.

## 3.2 Detailed study sites

We conducted detailed geomorphic study of two mountain ranges: the Talamanca Range of Costa Rica and the Central Range of Taiwan. The most recent research has converged on the hypothesis that high elevations were attained in both ranges after a rapid Plio-Pleistocene acceleration in rock uplift rate (Morell et al., 2012; Zeumann and Hampel, 2017; Hsu et al., 2016). This explanation leaves room for only incidental glacial

erosion, despite the long history of glacial geomorphic study in both places (e.g., Kano, 1932; Kano, 1935; Panzer, 1935; Weyl, 1955). A similar case can be made for other tropical mountain ranges, such as the Finisterre Range of Papua New Guinea, where proposed patterns of landscape evolution have not considered the influence glacial erosion (Abbott et al., 1997; Hovius et al., 1998), despite early—but not widely recognized—reports of glacial remnants there (e.g., Loeffler, 1971). We ultimately narrowed our foci to the

Talamanca Range and Central Range—both because of the extensive prior studies of their geodynamics and glacial geomorphology, and because of the apparent disconnect between these two lines of research. In the Talamanca Range, despite extensive geomorphic mapping carried out over several decades (Weyl, 1955; Hastenrath, 1973; Bergoeing, 1978; Barquero and Ellenberg, 1986; Shimizu, 1992; Orvis and Horn, 2000; Lachniet and Seltzer, 2002), the glacial chronology has been poorly constrained (Orvis and Horn, 2000); to

address this issue, we targeted Cerro Chirripó for [10]Be surface-exposure-age dating. In contrast, the glacial chronology of the Central Range is reasonably well constrained by such methods (Siame et al., 2007; Hebenstreit et al., 2011).

### 3.2.1 Talamanca Range, Costa Rica

The Talamanca Range is a high section of the Central American Volcanic Arc that stretches for ~175 km from central Costa Rica to western Panama (Fig. 1A). The range largely comprises Miocene volcanics and intermediate plutonics that intruded volcanic rocks around 8 Ma (Drummond et al., 1995) and cooled to <65°C by 5 Ma (Morell et al., 2012). The central Talamanca coincides with the subduction of the aseismic

Cocos Ridge, one of the most striking features of the Central American convergent margin. Subduction of the Cocos Ridge is thought to have contributed to the onset of rapid rock uplift, the development of a bivergent wedge, and the cessation of arc volcanism in the Talamanca (Morell et al., 2012). Recently, several authors have converged on the conclusion that Cocos Ridge subduction initiated sometime after 3 Ma, and that the extinct arc has been uplifted by a minimum of ~2 km in this time (Morell et al., 2012; Zeumann and

Hampel, 2017).

Several studies have attempted to link the erosional history of the Talamanca Range to the onset of Cocos Ridge subduction (Morell et al., 2012; Zeumann and Hampel, 2017). Significant disequilibrium observed in Talamanca river networks is thought to record a switch to an (ongoing) higher rate of rock uplift during the last 3 Myr. Zones of anomalously low-relief topography found at moderately high elevations (between 2000-3000 m), which are thought to represent an eroded surface that has been advected to its present-day elevation, have been cited as further evidence of a Plio-Pleistocene switch in rock uplift rate (Morell et al., 2012). Importantly, this prior work has excluded glacial erosion as a factor in the long-term evolution of the mountain range. Our work does not address the interpretation that low-relief landscapes found between 2000-3000 m elevation are part of an uplifted, eroded landscape, and neither do our results contradict the claim that a major shift in the rate of rock uplift has occurred recently in the Talamanca. Rather, we make a case that sufficient rock mass existed above the ELA for glaciation to occur during LGM, and possibly during cold stages prior to the LGM.

The highest landscape of the Talamanca Range is the Chirripó massif, a low-relief terrain spanning an area of ~75 km$^2$, perched above ~3000 m above sea-level, and surrounded by rugged, high-relief, ridge-and-valley, fluvially-driven topography. Glacial landforms—such as lateral moraines, glacially striated bedrock, roches moutonnées, over-deepened lakes, cirques and U-shaped valleys—were first reported on Chirripó in the 1950s and have been studied episodically since then (Weyl, 1955; Barquero and Ellenberg, 1986; Bergoeing, 1978; Hastenrath, 1973; Shimizu, 1992; Lachniet and Seltzer, 2002; Orvis and Horn, 2000). The most prominent cirques cut into the Cerro Chirripó peak, but smaller cirques are also scattered around the massif. Lateral moraines have been mapped at elevations as low as ~3150 m and as high as 3450 m in Valle de las Morrenas and Valle Talari, and hummocky recessional moraines can be found on cirque floors as high as 3500 m.

Orvis and Horn (2000) provided the first rigorous ELA estimate for maximum ice extent at Chirripó. They suggested an ELA of ~3500 m, based on standard ice surface reconstruction and a combination of balance ratio (BR) and accumulation area ratio (AAR) methods. This estimate corroborates earlier estimates by Weyl (1955) and Hastenrath (1973), who also suggested an ELA of 3500 m based on the elevation of cirque floors. Lachniet & Seltzer (2002) independently estimated a similar ELA using both AAR methods and the maximum elevation of lateral moraines. Although multiple authors have converged on a consistent ELA estimate using a variety of methods, the only age constraints for glacial timing available are minimum-limiting bulk [14]C dating of organic material in postglacial lakes (Orvis and Horn, 2000). Today, Costa Rica contains no glaciers, and snow has never been reported even at the highest elevations. Prior studies also

inferred a lack of glacial activity during the Holocene.

### 3.2.2 Central Range, Taiwan

The Central Range of Taiwan is the product of the oblique collision of the Luzon Arc and Eurasia, and it is comprised of metamorphosed marine sediments and pre-Cenozoic basement (Suppe, 1981). Due to the oblique nature of collision, it is thought that deformation has propagated from north to south for 5-7 Ma (Byrne and Liu, 2002). The northern 150 km of the Central Range have long been considered a type example of a fluvially-driven, steady state mountain belt (Willett and Brandon, 2002; Stolar et al., 2007). Recent work has countered this idea by putting forth the case that a rapid increase in exhumation rate occurred along the entire strike of the range starting between 1-2 Ma (Hsu et al., 2016). Low-relief surfaces found at high elevations (2800-3000 m) in Taiwan are thought to be remnants of an eroded surface formed sometime prior to 1-2 Ma (Ouimet et al, 2015).

Despite contemporary skepticism that glacial erosion has affected Taiwan, evidence supporting this contention have a long and well-established history, with the first observations dating back to the early 20[th] century (Kano, 1932; Kano, 1935; Panzer, 1935). Several groups have reported glacial remnants near the highest peaks of Taiwan's Central Range (Chu et al., 2000; Cui et al., 2002; Hebenstreit and Böse, 2003; Böse, 2004; Hebenstreit et al., 2006; Ono et al., 2005; Carcaillet et al., 2007; Siame et al., 2007; Hebenstreit et al., 2011) which lie within ~500 m of the estimated LGM ELA of ~3400 m (Hebenstreit et al., 2011). Glacial erosion features have been reported in three separate massifs, including Nanhudashan (also romanized as Nanhutashan; rendered as Nankotaisan in Japanese by Kano; Hebenstreit and Böse, 2006; Carcaillet et al., 2007; Siame et al., 2007; Hebenstreit et al., 2011), Xueshan (also romanized as Sheshan, Hsueshan, Hsuehshan, etc.; rendered as Tsugitakayama in the early literature; Cui et al., 2002) and Yushan (rendered in Japanese as Niitakayama in the early literature; Böse, 2004; Hebenstreit, 2006). The best preserved of these remnants are found at Nanhudashan and Xueshan, and include recessional moraines, polished (striated) bedrock, erratics, and cirques (Cui et al., 2002; Hebenstreit and Böse, 2004; Siame et al., 2007). Carcaillet et al., 2007, Siame et al. (2007), and Hebenstreit et al. (2011) carried out [10]Be analysis of scoured bedrock and boulders perched on moraines at Nanhudashan and found relatively young (15-9 ka) glacier retreat ages. At Xueshan, Cui et al. (2002) sampled moraines between 3300-3500 m for optically stimulated luminescence (OSL) and reported exposure ages of 14-44 ka. The timing of the local LGM is thus rather uncertain in Taiwan, and glacial ice appears to have persisted in some places as late as the Holocene.

ELA estimation in Taiwan has been a challenge for a number of reasons. One reason is that fluvial erosion of glaciated landscapes has been severe in places, which makes the maximum extent of glaciation in Taiwan

difficult to ascertain. Previous work has indicated the presence of a glacial diamict at 2250 m in a valley flanking Nanhudashan, far below unambiguous glacial valleys (Hebenstreit and Böse, 2006). Using the relict configuration of glaciated valleys, Hebenstreit (2006) estimated an ELA of 3355 m at Nanhudashan, specifically employing the terminal-to-summit altitudinal method (TSAM). Hebenstreit (2006) also used TSAM to estimate an undated ELA of 3400 m at Yushan, a third glaciated massif in southwest Taiwan. Other work has used the maximum vertical extent of lateral moraines in both Xueshan (Cui et al., 2002) and at Yushan (Böse, 2004) to propose an ELA of ~3400 m.

## 4 Data

The principal datasets used in this study are digital topography, high-resolution satellite imagery, and rock samples collected from Cerro Chirripó for $^{10}$Be exposure age analysis. All topographic analysis was performed on 1-arcsecond (projected at 30 m resolution) Shuttle Radar Topography Mission (SRTM) digital topography (Farr et al., 2007). SRTM data were acquired from the U.S. Geological Survey Earth Explorer website (available at https://earthexplorer.usgs.gov). Worldview-2 imagery (50 cm resolution) of Cerro Chirripó, Costa Rica taken in March 2012 was used to aid field mapping. Rock samples for $^{10}$Be exposure age analysis were collected at Cerro Chirripó in June 2014.

# 5 Methods

## 5.1 DEM processing

We generated a contiguous DEM of each tropical mountain range by mosaicking tiles of 1-arcsecond SRTM digital topography. Voids were patched with a 3-arcsecond void-filled SRTM DEM. Each DEM was projected using a local Lambert azimuthal equal-area projection. Pit filling and drainage delineation were then performed on each DEM using the TopoToolbox package in Matlab (Schwanghart and Kuhn, 2010; Schwanghart and Scherler, 2014).

## 5.2  Hypsometry

Hypsometry refers to the frequency distribution of elevation. With normalization, it becomes the probability distribution of elevation (also known as the altitude-area distribution: e.g., Strahler, 1952; Mitchell and
Montgomery, 2006). It has traditionally been represented as a "hypsometric integral", which is equivalent to a cumulative distribution or CDF (Strahler, 1952; Montgomery et al., 2001). When represented as a probability density function or PDF (e.g., Egholm et al., 2009), it is typically computed as a histogram. Here we instead use kernel-density estimation to compute the elevation PDF. This standard technique in statistics takes account of sampling (counting) uncertainty and generates a smooth function that is easier to interpret
Specifically, hypsometry is calculated as follows:

$$p(h) = \frac{1}{nw}\sum_{i=1}^{n} K\left(\frac{h-h_i}{w}\right) \tag{1}$$

where the frequency distribution of elevation $h$ is estimated by summing component smoothing functions of
form $K$ (here a Gaussian) and bandwidth $w$ (a function of the sample standard deviation) centered at sample elevations $h_i$.

### 5.2.1   Range-scale hypsometry

Hypsometry has been a popular but rather blunt instrument for assessing the influence of glacial erosion on landscapes. The advent of SRTM digital topography, first at 3-arcsecond and then at 1-arcsecond resolution, has facilitated the global deployment of hypsometry; it has also entailed the need to set a meaningful scale at which to segment the data. One approach (Egholm et al., (2009)) is to compute the elevation distribution for each 1°x1° tile, which has the advantages of simplicity and objectivity. It has the disadvantage that tile
boundaries often segment mountain ranges across major drainage divides, which sometimes results in spurious hypsometric maxima. To address this problem, we clipped the DEM of each mountain range by

manually tracing a bounding polygon along each range front, ensuring that both flanks of each range were included in the domain of analysis, and calculated the hypsometry of each.

### 5.2.2 Progressive hypsometry

We designed an algorithm that concisely describes how hypsometry varies with the scale of analysis. Development of this algorithm was motivated by the observation that the hypsometric maximum of large regions can shift by several kilometers depending on the boundaries of analysis. The algorithm, which we term "progressive hypsometry," (PH) involves the measurement of hypsometric maxima in nested catchments whose outlets span from the lowest to the highest elevations in a mountain range. Progressive hypsometry consists of three major components: (i) segmentation of the landscape into large catchments, (ii) calculation of hypsometry along flow paths, (iii) segmentation into nested subcatchments characterized by a shared modal elevation. We first segment the targeted mountain range into large (1000 km$^2$) catchments, hereafter referred to as "supercatchments", delineated on the condition that they link the main divide to a low reference elevation. This method typically segments each mountain range into 30-60 supercatchments. We then do the following:

1.  Map channel network:
    a.  define a channel network in each supercatchment using an arbitrary flow accumulation area threshold $A\_c$—this thins the set of all possible flow paths
    b.  traverse downstream from each channel head $i=1...N$ to the catchment exit to define a set of $N$ along-channel pixel chains
    c.  extend each chain $i$ upstream from its channel head to the drainage divide by following path of greatest flow accumulation area, ensuring that each pixel chain spans the full range of elevation from ridge to exit
2.  Map PH along network (Fig. 4):
    a.  traverse each chain $i$ upstream from the exit (shared by all chains)
    b.  map along each chain a nested series of subcatchments, one at every channel pixel $j(i)$
    c.  for each nested subcatchment, estimate its elevation pdf, its modal elevation $h\_mode\_j$ (where the pdf peaks) and its outlet elevation $h\_out\_j$ (Fig. 3)
    d.  record as a set of $i=1...N$ sequences of $[h\_out\_j(i),h\_mode\_j(i)]$ pairs
3.  Identify all PH "benches", characteristic nested-catchment modal elevations (Fig. 5)
    a.  perform change-point detection along each chain $i=1...N$ to locate and define large jumps in $h\_mode$ at each $h\_out$

b.  define the outlet elevation *h_out* at each jump as *h_change*

c.  designate the groups of between-jump modal elevations *{h_mode}* as "benches"

d.  define each bench modal elevation *h_bench = min{h_mode}*

e.  record as a set of *i=1…N* sequences (one per chain) of *[h_change_k(i),h_bench_k(i)]* pairs, each of length *k(i)=1..n(i)*

f.  concatenate all *N* sequences of *[h_change_k(i),h_bench_k(i)]*

We performed progressive hypsometry on the ten selected mountain belts, using a low-elevation reference level of 150-250 m in each mountain range. This reference elevation focuses the analysis just above large depositional plains, which improves the efficiency of the algorithm. The Rwenzori are an exception, since they are more than 1000 km from the nearest coast and rise sharply above lowlands with several large lakes at 1200 m. We used a low-elevation reference level of 1200 m for the Rwenzori.

**5.3 Focus sites**

**5.3.1 Assessment of glacial and post-glacial morphology**

We compiled observations made during field campaigns at Cerro Chirripó (in 2014, 2016) and Nanhudashan (in 2015), satellite/aerial imagery, and maps produced by previous workers to map glacial landforms and estimate maximum ice extent in both the Talamanca Range and Central Range. Glacial landscapes are only present at Cerro Chirripó in the Talamanca Range; in Taiwan, glacial remnants are best preserved at Nanhudashan: we focused on these two massifs. We also selected glaciated catchments to compare glacial geomorphic mapping with detailed hypsometric analysis. Glacial advances in both focus sites terminated near 3000 m, so for ease of comparison all glaciated catchments were extracted and delineated (in a GIS) using a common 3000 m outlet elevation.

Glacial valleys at Chirripó generally have a much lower gradient than the fluvial valleys flanking them, and the very presence of well-preserved glacial landforms, such as sharp-crested moraines, together with the apparent absence of Holocene landsliding, indicates that post-glacial erosion has been slow in these glaciated zones. In contrast, signs of relatively fast erosion, such as frequent landsliding, are apparent in the surrounding fluvial valleys in satellite imagery and air photographs, and were confirmed in the field. There have been no direct measurements of erosion rate in these fluvial catchments, but it is thought that a fluvial erosion rate of ~1 mmyr$^{-1}$ has been sustained in parts of Costa Rica for >2 Myr (Morrell, et al., 2012). The boundary between the fluvial and glacial domain in these landscapes usually gives rise to an erosion front: that is, a pronounced topographic break between the slowly eroding, relatively low-sloping glacial valleys and steep, quickly eroding fluvial valleys. We mapped in detail the erosion fronts in both places. This

mapping was guided by sharp changes in slope, as well as by the abrupt disappearance of glacial deposits and the transition to non-glaciated bedrock cliff faces (Fig. 6). Delineation of erosion fronts was qualitative and subjective, since no objective metrics are available for distinguishing perched glaciated valleys from headward-propagating fluvial valleys. Fluvial-based metrics such as normalized channel steepness are not particularly useful for our purposes, since the effects of significant glaciation have altered the landscape beyond any meaningful application.

To guide our qualitative assessment, we developed a set of rules. First, we used a binary slope map (with a threshold of 35°) to identify places where low-sloping glacial valley floors made a hard transition to a fluvial-linked escarpment. Where these escarpments where linked to amphitheater heads, we mapped the entire amphitheater head; the initial roughness of erosion front boundaries was thus set by the 30 m resolution DEM and not by the sub-meter resolution imagery. Next, we used the imagery to check that all mapped erosion fronts coincided with the disappearance of glacial features, or with clear signs of ongoing erosion such as multiple, recent landslide scars. Finally, we excluded mapped zones that appear to be related to isolated events, such as single landslide scars that are not unambiguously linked to the ongoing propagation of the fluvial network into glaciated terrain.

To quantitatively describe the pattern of fluvial scarp encroachment into glaciated terrain, we define two new metrics. The first, which we term the ELA-Relative Modal Elevation (ERME), measures the difference between hypsometric maxima of glaciated catchments and the estimated LGM ELA. At both Cerro Chirripó and Nanhudashan, this metric is calculated using the local estimate of the LGM ELA. Both massifs are small enough that large climatic gradients are not likely to drive substantial differences in the position of the ELA, as has been documented in much larger tropical mountain ranges (Prentice et al., 2005; Stansell et al., 2007). We hypothesized that the duration and intensity of glacial erosion in all glaciated catchments was similar during the LGM, and that glacial erosion would thus leave a characteristic modal elevation at the ELA in all glaciated catchments. We further hypothesized that fluvial scarp encroachment of glaciated terrain would bias hypsometric maxima at the ELA to higher elevations.

The second metric, which we call the Scarp Encroachment Ratio (SER), is an approximation of the headward distance traveled by the fluvial escarpment into each glaciated catchment (Fig. 6). This distance is expressed as a ratio of scarp-affected terrain to all terrain in glaciated catchments. In each glaciated catchment, we found the area below and above the escarpment, $A_c$ and $A_g$, respectively, and calculated a corresponding length scale for each: $L_c$ and $L_g$. SER is the relative length scale:

$$SER = \frac{L_c}{L_c + L_g} \tag{2}$$

At its core, SER is a quantification of the qualitative observation that fluvial erosion encroaches (destroys) glaciated terrain. At peak glacial conditions, both glacial and fluvial erosion would have been ineffective near the glacial terminus, since glacial erosion converges towards zero near this elevation and simultaneously blocks fluvial incision. To measure the elevation gain of the post-glacial scarp, we assume each scarp originated near the LGM glacial terminus. We thus choose to compare SER in glaciated catchments above the 3000 m benchmark.

### 5.3.2  Surface-exposure age dating of deglaciation

Six samples were collected for $^{10}$Be exposure dating from boulders embedded in both lateral and frontal recessional moraines at 3400–3500 m elevation in Valle de las Morrenas and Valle Talari, two samples from scoured bedrock within ~15 m of the Chirripó summit, and one sample from a landslide boulder sourced from a cirque headwall (Fig. 7). Processing at Lamont-Doherty Earth Observatory and measurement at Lawrence Livermore National Laboratory followed standard procedures (e.g., Schaefer et al., 2009), and $^{10}$Be ages were calculated with the CRONUS-Earth online calculator (Balco et al., 2008) v.2.2, using a low latitude, high elevation production rate obtained in Peru by Kelly et al. (2013) and the scaling scheme of Lal (1991) and Stone (2009) (Table S1, S2).

# 6  Results

## 6.1  Range scale hypsometry

In Fig. 8a2-j2 we compare the range-scale hypsometry of each of the selected mountain ranges to the tropics-wide cpELA band, 3400-4000 m. Most of the selected ranges show that the fractional area occupied by each elevation band decreases steadily with increasing altitude, although in some cases, such as the Finisterre Range (Fig. 8e2), topographic plateaus are evident at high elevations and are recorded as secondary hypsometric maxima. In some cases, the highest elevations fall within the bounds of the tropical cpELA, and in other cases peaks extend far above it, but none of these mountain ranges has a significant hypsometric maximum at or above the cpELA. In other words, each range was high enough for cold-phase glacial erosion (with the exception of the Leuser Range), and yet the extent and prevalence of glaciated landscapes in all of them appears small relative to the size of fluvial landscapes.

## 6.2 Progressive hypsometry

Nine of the ten selected mountain ranges have catchments with a PH modal elevation at the cpELA, which we consider a record of those catchments having established a perched glacial base-level (Fig. 8b1-j1). Variability in glacial influence between each range is best assessed on the basis of paired outlet and modal elevations. For example, catchments in some ranges have an outlet elevation as low as 150 m and a modal elevation between the cpELA and wpELA. In these mountain ranges, the progressive hypsometry of the largest catchments is thus dominated by glacial erosion. In Fig. 8f1-j1, these large catchments are represented by a left spread of points within the ELA band. In other ranges, catchments with a modal elevation within the bounds of the cpELA are only found above ~2000 m. Only the Leuser Range shows no apparent signs of glacial action.

These results stand in contrast to range-scale hypsometry, which indicates minor glacial influence across all of the selected mountain ranges. Progressive hypsometry instead highlights variability in the prevalence of glaciated terrain and points to glacial erosion having had an influence in most of these mountain ranges, and a particularly significant role in some. Our two focus sites in Costa Rica and Taiwan are examples of the former—where glacial erosion has apparently had only a marginal influence.

## 6.3 Focus site #1: Cerro Chirripó, Talamanca Range, Costa Rica

Classic examples of glacial landforms are found at Cerro Chirripó (Fig. 9). Several valleys host kilometer-scale lateral moraines and most valleys are blanketed by glacial till and recessional moraines. Striated and

scoured bedrock is prevalent in the paleo-accumulation zone in the major valleys emanating from the Chirripó peak. Most glaciated catchments at Chirripó have a modal elevation within several meters of the estimated local LGM ELA of 3500 m (Fig. 11C). Two high catchments have a modal elevation that is somewhat higher than the local ELA. An important feature is that these catchments appear to be heavily modified by scarp

encroachment, which leads us to infer that fluvial scarp encroachment has erased the lower portion of their glacially eroded topography and has biased their hypsometric maxima to higher elevations. Field observations corroborated this inference (e.g., Fig. 6c-e).

Our [10]Be ages tie the glacial Chirripó landscape to the LGM and provide the first constraints on its termination

in Costa Rica (Fig. 9A). Lateral and recessional moraine boulders yielded ages between $18.3 \pm 0.5$ ka and $16.9 \pm 0.5$ ka, and in Valle de las Morrenas ages tend to young toward the headwall. Near the Chirripó summit, a bedrock surface gave an age of $22.0 \pm 0.7$ ka. This age may reflect thinning of the ice prior to ~18 ka if there is no inherited [10]Be in the sample. A landslide boulder sourced from a cirque headwall and deposited above moraines gave an age of $15.2 \pm 0.5$ ka. Rock avalanches and landslides sourced from steep

valley headwalls are a common feature in most post-glacial landscapes (Ballantyne, 2002; McColl, 2012; Ballantyne, 2013). The fact that such deposits are not advected down-valley by flowing ice provides a useful constraint on the onset of post-glacial conditions. The other bedrock sample yielded an age of $8.9 \pm 0.4$ ka. We infer that this surface has a substantially younger exposure age than the LGM due to burial by soil or sediment.

## 6.4 Focus site #2: Nanhudashan, Central Range, Taiwan

Glacial landforms are not preserved as clearly at Nanhudashan (Fig. 10) as they are at Chirripó, although scoured bedrock and recessional moraines have been mapped and dated to the last glacial (Siame et al. 2007;

Hebenstreit et al., 2011). Perhaps the most striking feature of the glacial landscape at Nanhudashan is its asymmetry. The south-eastern glacial valley of Nanhudashan is about 2 km long and has the best preserved glacial remnants. A small plateau that fed north- flowing glacial ice has been largely removed by scarp encroachment from the south-west. Recessional moraines in the north-western glacial valley disappear abruptly below and erosion front just below 3400 m. In the south-west, almost no glacial remnants remain.

Glaciated catchments at Nanhudashan have a hypsometric maximum at or above the estimated LGM ELA of 3400 m. In the best preserved glacial valley (southeast), the hypsometric maximum is within 35 m of the ELA (Fig. 11D). The hypsometric maxima of the other two glaciated catchments are more than 100 m above the estimated LGM ELA (Fig. 11D). Scarp encroachment has been far more severe in these valleys.

**6.5 Comparison of focus site landscapes**

ERME and SER reveal similar patterns of glacial erosion and scarp encroachment at Chirripó and Nanhudashan, only scarp encroachment is more advanced at Nanhudashan (Fig. 12). Of the three glacial catchments at Nanhudashan, two show greater scarp encroachment than any at Chirripó, a pattern reflected in their ERME and SER values. The escarpments propagating into glacially eroded terrain at Nanhudashan, some of which were mapped in previous studies (e.g., Hebenstreit et al., 2006; Willett et al., 2014), are particularly spectacular. When combined, our observations at Chirripó and Nanhudashan capture a continuum of scarp encroachment into glacially eroded landscapes and the alteration of glacial-type hypsometry.

# 7 Discussion

## 7.1 Hypsometry

Our hypsometric analyses reveal signs of glacial limitation recorded in the topography of multiple tropical mountain ranges. While application of traditional hypsometry provides hints of glacial influence, the new tool of progressive hypsometry amplifies its topographic signal and exposes evidence that glacial erosion has had a profound effect on the height of these mountain ranges. This inference was tested at focus sites in Taiwan and Costa Rica, expressly chosen because their exposure to glacial erosion has been apparently weak

to negligible. Even in these seemingly marginal examples, we find substantial evidence that valleys with a modal elevation near the ELA originate in glacial erosion. Our observations support the claim that a glacial base-level has been periodically introduced in mountain ranges throughout the tropics.

    Glacial limitation is expressed differently in tropical mountains than in mid-latitude mountains, because the

high cpELA imposes a fundamentally different relationship between tropical glaciers and their flanking fluvial systems. At mid-latitudes, the cpELA descends to ~2000 m or lower (Broecker and Denton, 1989; Egholm et al., 2009), and cirque-fed valley glaciers have commonly extended to fluvial base-level (and sometimes to sea-level) during cold-phase glacial advances. In the tropics, ice flux of this proportion is only rarely observed (among our selected ranges, only in the Rwenzori). Instead, in the tropics we find glacial

landscapes that sit perched above fluvial valleys. These glacial valleys are responsive to a high elevation base-level at the ELA and are disconnected from fluvial base-level control.

    The long-term evolution of the high-elevation, glacial base-level has proceeded differently between tropical mountain ranges, and for this reason it is important to consider the potential for glacial limitation in the

context of all of the mountains analyzed. Three factors determine the development and preservation potential of a glacial base-level in the tropics: (i) the volume and pattern of rock uplift through the cpELA; (ii) the efficacy of glacial erosion; and (iii) fluvially-driven destruction of glaciated terrain. Valleys are initially disconnected from fluvial base-level once enough rock mass has passed through the cpELA for glacier formation. If glacial erosion is strong enough during the initial phases of glaciation, it drives the development

of a modal elevation develops at the cpELA. During interglacials, fluvially-driven scarp encroachment removes portions of perched, glaciated valleys. If glaciated landscapes above the cpELA are spared scarp encroachment, continued rock uplift and subsequent cold-phase glacial erosion drives further development of the glacial base-level at the cpELA. If not, evidence of the perched glacial base-level is erased. Progressive hypsometry appears to capture a continuum of this behavior.

Landscapes can be thought of as a mosaic of catchments. Progressive hypsometry objectively distinguishes these catchments on the basis of their modal elevation, and in doing so offers a fine-scale perspective on the distribution of terrain (Fig. 8a1-j1). The pairing of modal elevation and catchment outlets also provides an indication of how catchment geometry varies with elevation. For example, catchments with a large vertical gap (Fig. 13c) between their modal elevation and outlet must occupy disproportionately broad areas at high elevations. This pattern reflects either perched, low-gradient terrain or erosional penetration and contour elongation at the modal elevation.

Headward glacial erosion both penetrates the landscape near the ELA and promotes the formation of low-gradient terrain. When mountains first pass through the cpELA, this effect is minor. If glacial erosion repeatedly acts on the same landscape through multiple glaciations, the relative strength of the modal elevation at the ELA increases. In Fig. 8a1-j1, we highlight those catchments with a (PH) modal elevation within the bounds of the cpELA and wpELA. In the Central Range of Taiwan, Talamanca Range, Finisterre Range, Crocker Range, and Owen Stanley Range, these glaciated catchments sit at the highest elevations, and are nested inside fluvial catchments (Fig. 8b1-f1). In the Merauke Range, Mérida Range, Sierra Nevada de Santa Marta, and the Rwenzori (Fig. 8g1-j1), rock mass has been advected high above the cpELA (reaching as high as 5700 m in Santa Marta). In these four mountain ranges, however, the largest vertical gap between catchment modal elevation and outlet are found at the cpELA.

Our data show that once a catchment reaches the cpELA, the bulk of its terrain does not rise much higher. The strongest evidence of this limitation phenomenon is found in Fig. 8f1-j1, where the (PH) modal elevation of the largest catchments sits between the cpELA and wpELA. Furthermore, we do not observe any significant catchment (PH) modal elevations above the wpELA, even though peak elevations often extend high above it. These observations are consistent with the introduction of a glacial base-level at the cpELA, with supra-ELA terrain ultimately tied to an expanding zone of cirque/valley glacier incision. Progressive hypsometry thus supports the idea that glacial limitation is a viable mechanism in some tropical mountain ranges.

The Leuser Range is the only range among those analyzed that bears no signal of glacial erosion, and so we consider its progressive hypsometry as a null reference model by which to compare the progressive hypsometry of other glaciated ranges. In Fig. 13, we vertically exaggerate the progressive hypsometry of the Leuser Range to schematically illustrate the evolution of a mountain range that is spared glacial limitation as it rises through the cpELA. In the two null examples (Figs. 13a,b), rock uplift alone drives the growth of landscapes above the ELA, or the long-term effect of glacial erosion is compromised by fluvial erosion during

interglacial periods. In fig. 13a, the range is high enough for glaciation, but glacial erosion is either absent or negligible, or largely erased by fluvial erosion. Similarly, in Fig. 13b, glacial erosion has been either been negligible or severely compromised, and no significant record of glacial erosion has been left, even though more rock mass has passed through the ELA. In contrast, in Fig. 13c, the process of glacial limitation focuses catchment growth at the ELA, and glacial landscapes are preserved enough to leave a strong signal in progressive hypsometry. In this case, catchments enlarge near the ELA as more rock mass is pushed through it, strengthening the modal elevation observed there. Only isolated spires of rock reach higher elevations.

Considered in isolation, most of the mountain ranges analyzed would not be classified as glacially-limited. Yet, across the tropics, the highest mountain ranges bear the strong appearance of glacial limitation (Table 1). We next explore the potential for glacial limitation is those mountain ranges where a glacial influence is present but weaker, namely, the Central Range of Taiwan, Finisterre Range, Talamanca Range, Crocker Range, and Owen Stanley Range. The size of these glacial landscapes is limited by either a dearth of rock mass that has been advected through the cpELA or by the fluvial destruction of glaciated terrain. These mountains are therefore not glacially limited in the same sense that the other selected mountain ranges are. Nevertheless, their highest elevations all coincide with cpELA.

## 7.2 Glacio-fluvial limitation of mountain height

We have presented evidence for the periodic introduction and subsequent removal of a glacial base-level in the Talamanca Range and Central Range of Taiwan. Both ranges were glacially eroded during the LGM; their glaciated catchments both have a modal elevation near the cpELA; and in both landscapes, post-glacial fluvially-driven erosion has erased significant fractions of their glaciated terrain. These observations are consistent with three possibilities:

(i) *Fluvial limitation at cpELA with glacial ornamentation:* each mountain range is close to a fluvially-limited, steady state elevation (Fig. 2a). Transient landscapes of unspecified origin periodically reach the cpELA, and are briefly occupied by geomorphically ineffective glaciers. Scarp encroachment of glaciated terrain is incidental.

(ii) *Glacio-fluvial limitation:* an unspecified volume of rock mass has been advected through the cpELA, and glacial erosion and fluvially-driven scarp encroachment have been sufficient to remove it, thus limiting mountain height to the cpELA (Figs. 15, 2c).

(iii) *Fluvial limitation at a higher elevation:* both ranges are in a state of transience and will continue to grow and steepen until fluvial-limitation is achieved far above the cpELA.

The question of which scenario best describes the Talamanca Range and Central Range hinges on whether glaciations have been a rare (or even isolated) occurrence in these places. Unfortunately, our evidence highlights barriers to answering this question. Specifically, glacial landscapes are prone to erasure in both places (Fig. 12). In Taiwan, erasure of glacial landscapes appears to progress rapidly, and they are unlikely to survive the long, ~100 kyr glacial-interglacial cycle. It is also likely that if pre-LGM glaciations have occurred in these landscapes, their remnants have been removed.

Given the pattern of glacial limitation observed throughout the tropics, it is possible that the absence of significant terrain at the cpELA in Costa Rica and Taiwan is due to particularly effective scarp encroachment rather than ineffective glacial erosion. Comparable mountain ranges in the tropics have all been subject to significant glacial erosion upon passage through the cpELA, and it would be a remarkable coincidence if the Talamanca Range and Central Range, as well as the Finisterre Range, Crocker Range, and Owen Stanley Range were all independently limited at the same elevation close to the cpELA. Furthermore, fluvial erosion has not limited rock mass to elevations below the cpELA in either Costa Rica or Taiwan. Moderately high elevation (2000-3000 m), low-relief landscapes in both ranges that have been disconnected from external base level are further evidence that fluvial limitation in general is unlikely (Morell et al., 2012; Ouimet et al., 2015). In other words, if fluvial erosion can limit the height of these mountain ranges, it would likely do so at an elevation above the cpELA, and would ultimately be subject to glacial limitation in the future. We therefore propose glacio-fluvial limitation as the most parsimonious explanation for the coincidence of high peaks and the cpELA in the Talamanca Range and Central Range of Taiwan (Fig. 14).

**7.3 Steady-state landscapes revisited**

Mountain ranges are thought to evolve towards a steady state ultimately determined by the efficacy of fluvial erosion. Steady state is assessed on the basis of material flux, in which accretion of rock and sediment are balanced by denudation, and topography, such that mountain range height and width are invariant over appropriate spatial and temporal averaging scales (Willet and Brandon, 2002). Thermal steady-state (time-invariant temperature field) is a prerequisite for flux steady-state (Willett and Brandon, 2002). For decades, the Central Range of Taiwan has been considered a steady state, fluvially-limited mountain range (Suppe, 1981; Willett & Brandon, 2002; Stolar et al., 2007) on the basis of its relatively uniform height and width (Stolar et al., 2007) and a correlation between long- and short-term denudation rates (Dadson et al., 2003).

It is difficult to reconcile the steady-state model with evidence for periodic glacial erosion at high elevations

in Taiwan. This phenomenon is equivalent to introducing an upper boundary condition on mountain growth, which violates the conditions required for fluvially-driven steady state; essentially, glacial erosion imposes an upper limit on river channel steepness. Furthermore, it would be a striking coincidence for the Central Range to have reached a fluvially-driven steady-state at the cpELA, an elevation at which a glacial base-level is apparent in eight other similar mountain ranges. We suggest that on the long-term, an oscillatory pattern of glacial and fluvial erosion maintains the Central Range at its regionally uniform height.

# 8 Conclusions

Evidence for glacial limitation is widespread in high tropical mountains and has largely been overlooked. One reason is that the greater relative extent of fluvial versus glacial landscapes obscures the effects of glacial erosion when carrying out traditional hypsometric analysis. A second reason is that fluvially-driven scarp encroachment can erase signs of glaciation. We have addressed both issues and have presented evidence that glacial erosion has had a profound effect on limiting range height in rapidly uplifting and eroding tropical mountain ranges. Even in the most marginally glaciated mountain ranges in the tropics, glacio-fluvial limitation may drive long-lived, cyclic competition between glacial erosion and fluvial scarp encroachment, thereby preventing the establishment of steady-state in the classical sense.

**Competing Interests**

The authors declare that they have no conflict of interest.

## 5 Acknowledgements

This research was supported by the Lamont Climate Center. We are grateful to the Área de Conservación La Amistad Pacífico of the Sistema Nacional de Áreas de Conservación of Costa Rica for permission to conduct research at Cerro Chirripó, and to Chirripó National Park for providing accommodations during fieldwork. Lionel Siame generously led M. Cunningham and C. P. Stark in the field at Nanhudashan.

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

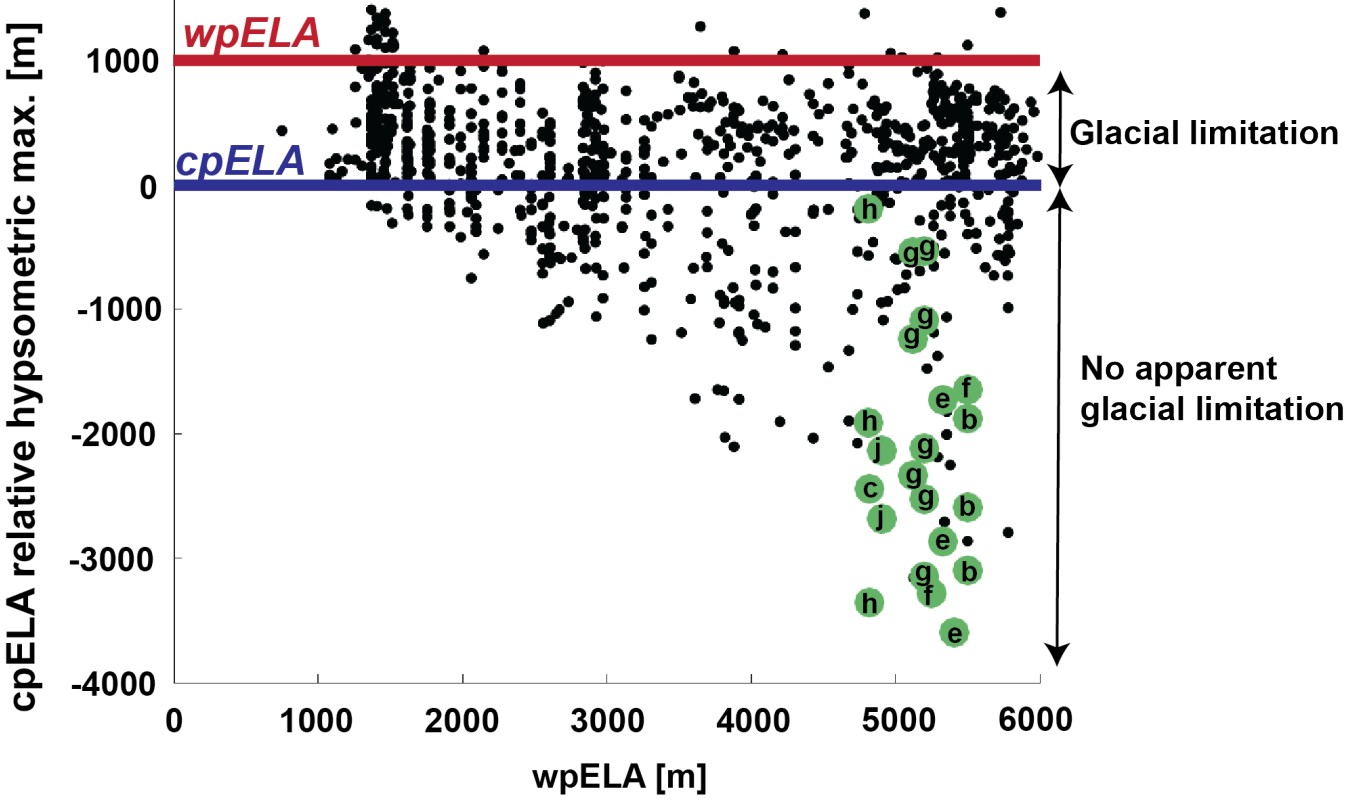

**Figure 1:** Hypsometric maximum of glaciated 1°x1° SRTM tiles (adapted from Fig. 2 of Egholm et al., 2009, data provided courtesy of V. Pedersen). Each tile is plotted by its approximate warm-phase (modern) ELA (wpELA; x-axis) and its hypsometric maximum relative to (after subtraction by) the cold-phase ELA (cpELA; y-axis); zero on the y-axis therefore indicates a hypsometric maximum at the cpELA. Glacial limitation is inferred for all SRTM tiles with a hypsometric maximum between the wpELA and cpELA. The tiles of tropical mountain ranges analyzed in this study are in green and labeled according to the scheme used in Figs. 3 and 8: (a) Leuser Range, Aceh (omitted here); (b) Central Range, Taiwan; (c) Talamanca Range, Costa Rica; (d) Crocker Range, Borneo (omitted here); (e) Finisterre Range, Papua New Guinea; (f) Owen Stanley Range, Papua New Guinea; (g) Merauke Range, Papua; (h) Mérida Range, Venezuela; (i) Sierra Nevada de Santa Marta, Colombia (omitted here); (j) Rwenzori, East Africa. The Leuser Range, Crocker Range, and Santa Marta were not included in the analysis by Egholm et al., 2009.

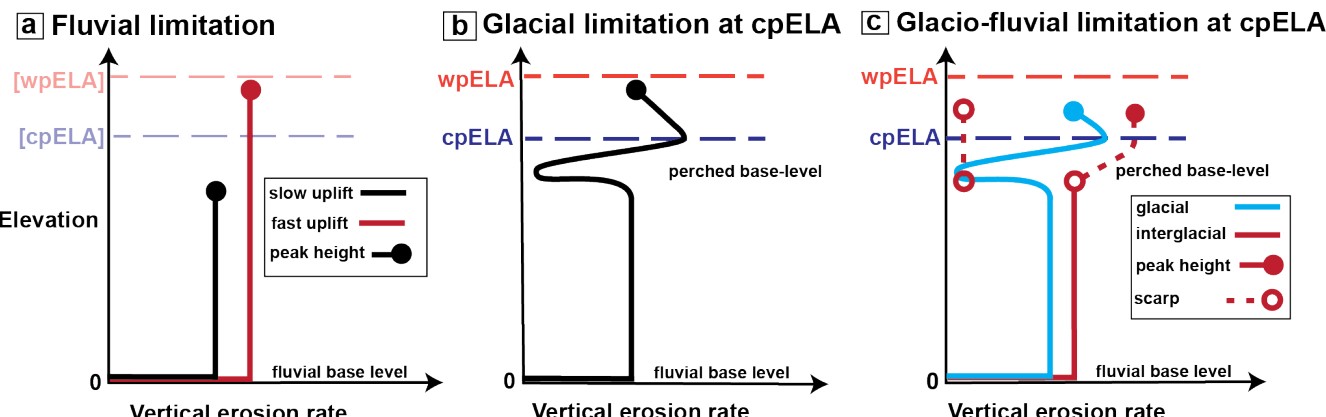

**Figure 2: (a)** Fluvial limitation: vertical erosion rate vs. elevation in steady state, fluvially-limited regime. Black line: slow rock uplift; red line: fast rock uplift. The rate of rock uplift sets steady-state peak elevations (closed circle) at different elevations. Warm-phase ELA (wpELA) and cold-phase ELA (cpELA) are indicated for reference, but are irrelevant in this scenario. **(b)** Glacial limitation, with glacial base-level below the cpELA. Black line: erosion rate profile, with significant glacial influence at high elevations. Peak elevations reach above the cpELA, but are tied to glacial incision near this elevation. **(c)** Glacio-fluvial limitation. Blue line: erosion rate profile during glacial periods, similar to (b). Red line: interglacial erosion rate profile, characterized by headward migrating escarpment (along dashed line above erosion rate profile). During interglacials, erosion in previously glaciated landscapes is ineffective (dashed line, left hand side).

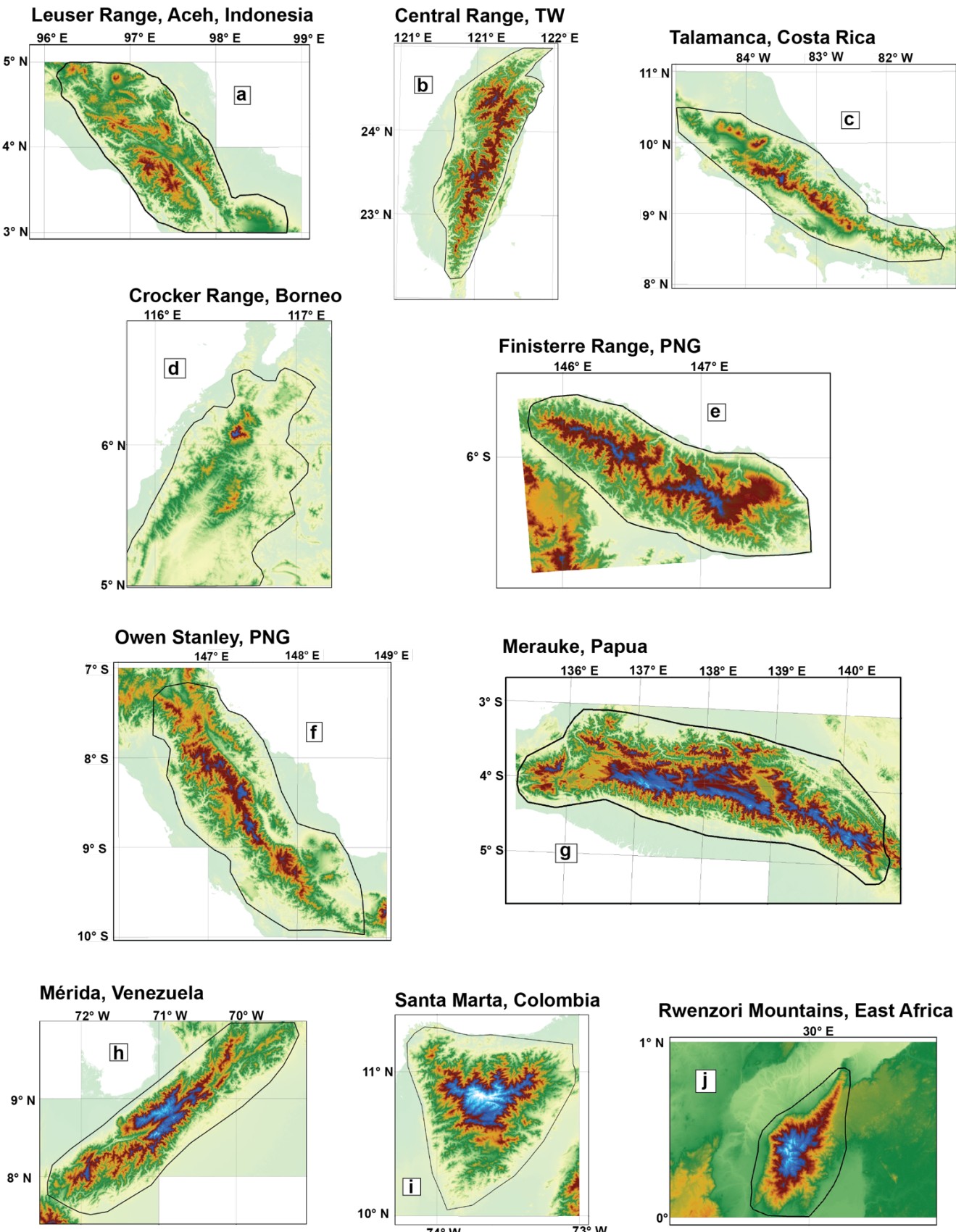

**Figure 3: (a-j)** SRTM DEM of selected tropical mountain belts. Yellow-green through red spans elevations 0-3400 m. Dark blue to light blue is 3400-4500 m (tropical cpELA to wpELA). Black polygons circumscribing each range indicate bounds for hypsometric analysis in Fig. 8.

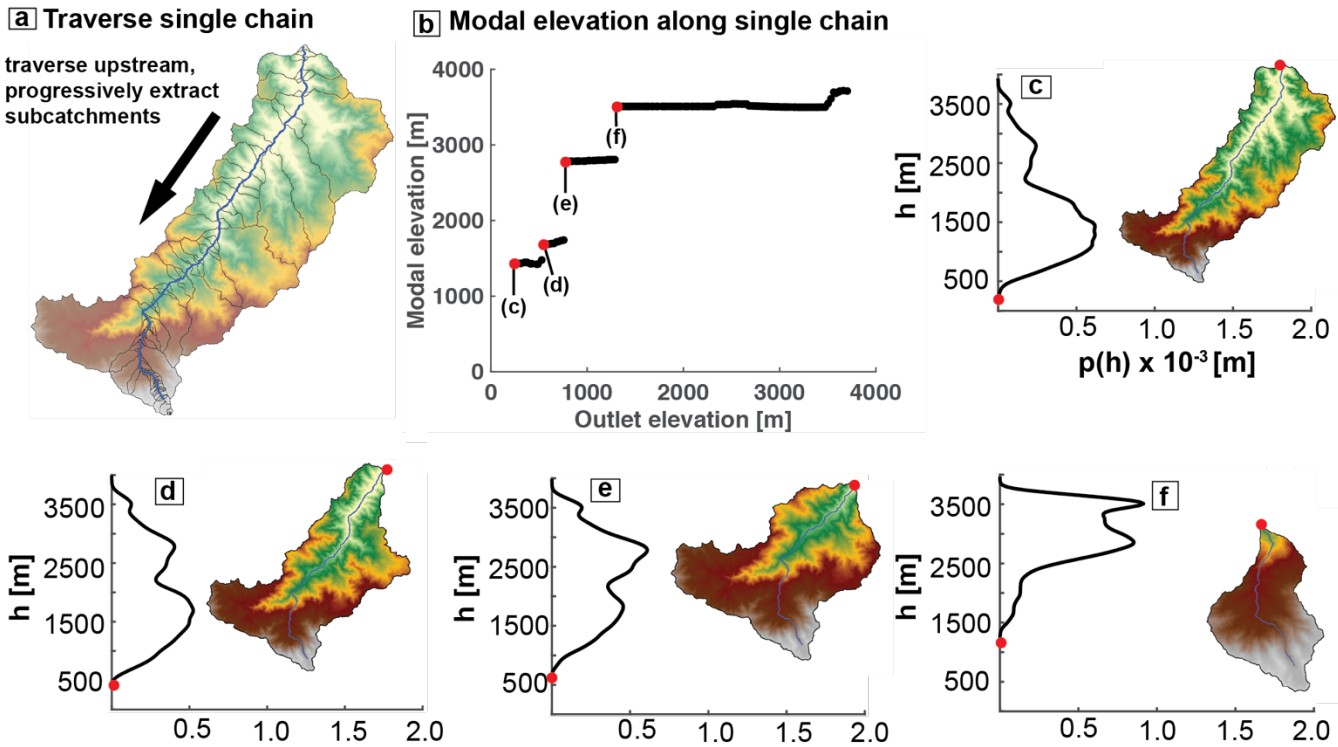

**Figure 4:** Progressive hypsometry in detail. **(a)** DEM of one supercatchment from the Talamanca Range. Elevation 0-4000 m is yellow through white. The dark blue streamline is an example of one chain along which progressive hypsometry is performed. The bounds of each progressively delineated catchment are drawn in black. **(b)** Modal elevation (hypsometric maximum) of catchments draining to progressively higher outlet elevations along dark blue streamline in (a). Each subcatchment in (a) is represented on (b). The stepped pattern in catchment modal elevation is commonly observed in all landscapes. **(c-f)** Elevation pdf and DEM of catchments at each jump in modal elevation in (b). Red dots indicate the catchment outlet on both the pdf and DEM, and are also represented as red dots in (b).

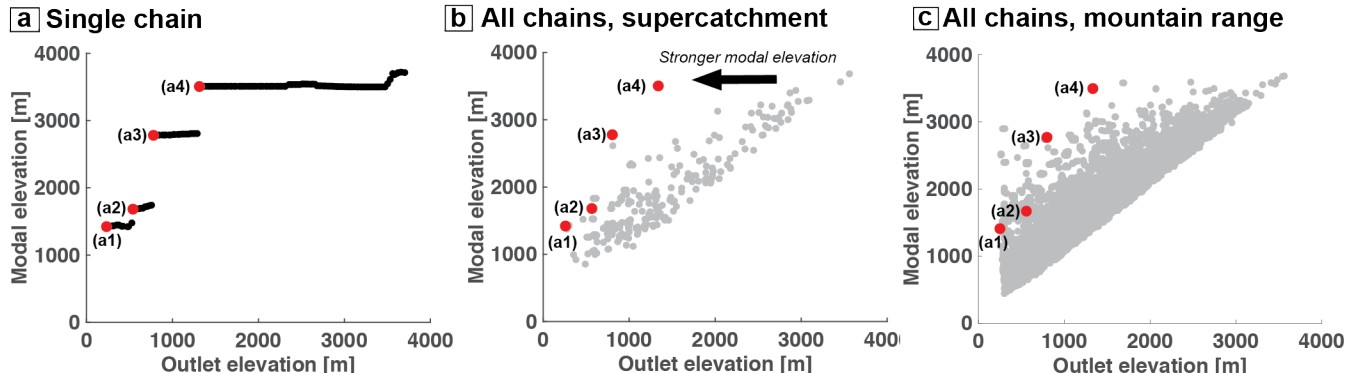

**Figure 5:** Mapping single progressive hypsometry chains on the mountain range scale. **(a)** Same as Fig. 4b. Modal elevation is calculated for subcatchments progressively extracted along single chain. **(b)** On the scale of a supercatchment (e.g., 4a), jumps in modal elevation for each chain are plotted corresponding to the outlet elevation at which the jump occurs (*h_change*). Red points are same as (a) and correspond with catchments in Fig. 4 (c-f). **(c)** Same as **b**, except for all chains in entire mountain range (Talamanca Range, Costa Rica).

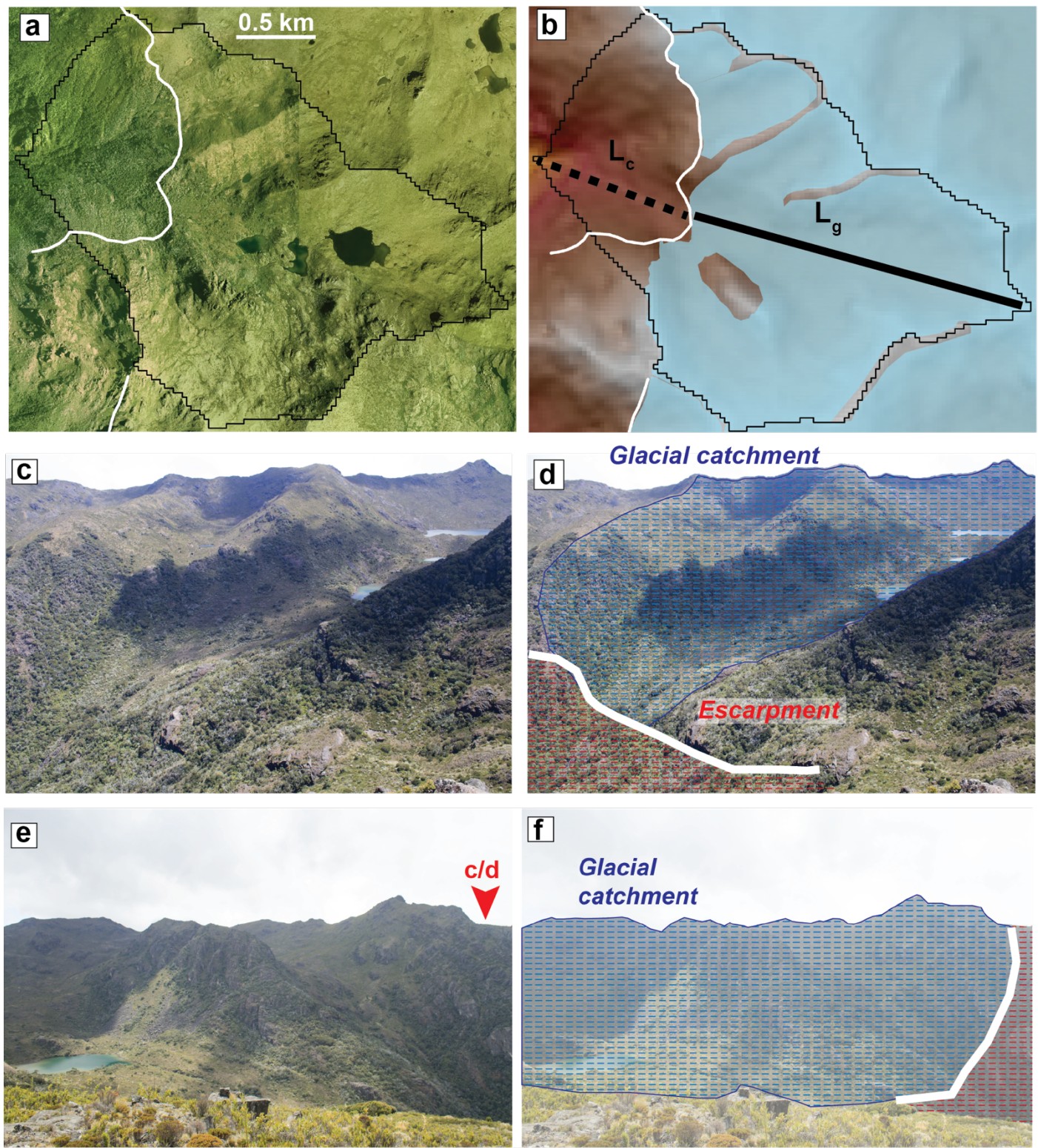

**Figure 6:** Scarp encroachment ratio (SER) calculation. **(a)** Aerial image of Valle de los Lagos, Chirripó, Costa Rica. Mapped escarpment in white. Boundary of catchment draining to 3000 m outlet in black. **(b)** Mapped LGM ice extent in light blue draped over DEM of Valle de los Lagos (same bounds as in (a)). Length scales $L_c$ and $L_g$ correspond to area below and above escarpment, respectively. SER calculation is presented in Eq. 2. **(c-f)**: Field photos of mapped escarpment. Blue and red zones in **d and f** correspond to glaciated zone and escarpment. White line is the same as in (a-b). **(e)**: Vantage point for photo **c/d** labeled with red arrow.

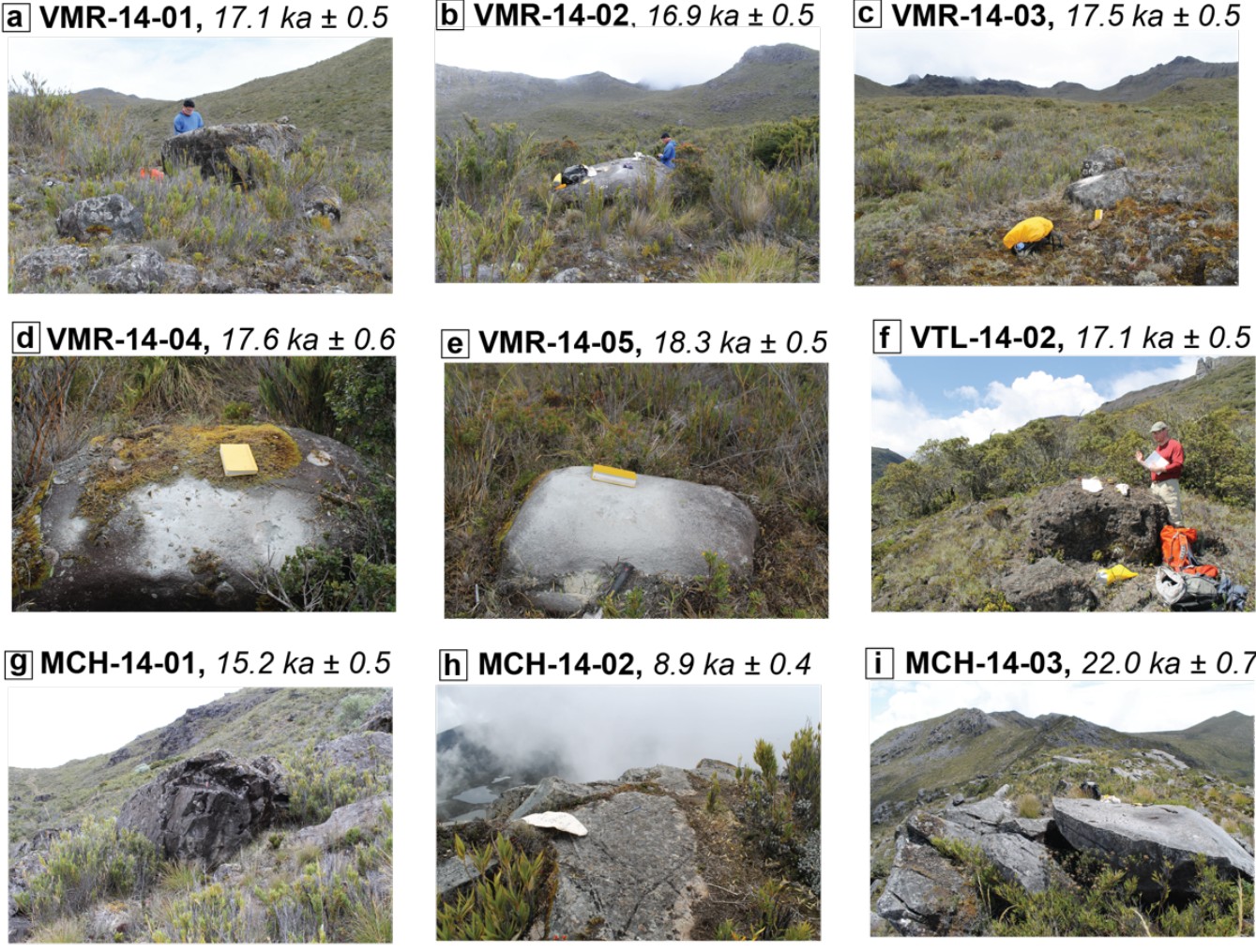

**Figure 7:** [10]Be sample locations for Cerro Chirripó. **(a-e)** Boulders (diorite) perched on recessional moraines in Valle de las Morrenas. **(f)** Boulder (andesite with quartz veins) perched on lateral moraine in Valle Talari. **(g)** Post-glacial landslide boulder. **(h-i)** Scoured bedrock along divide separating Valle de las Morrenas from Valle de los Lagos.

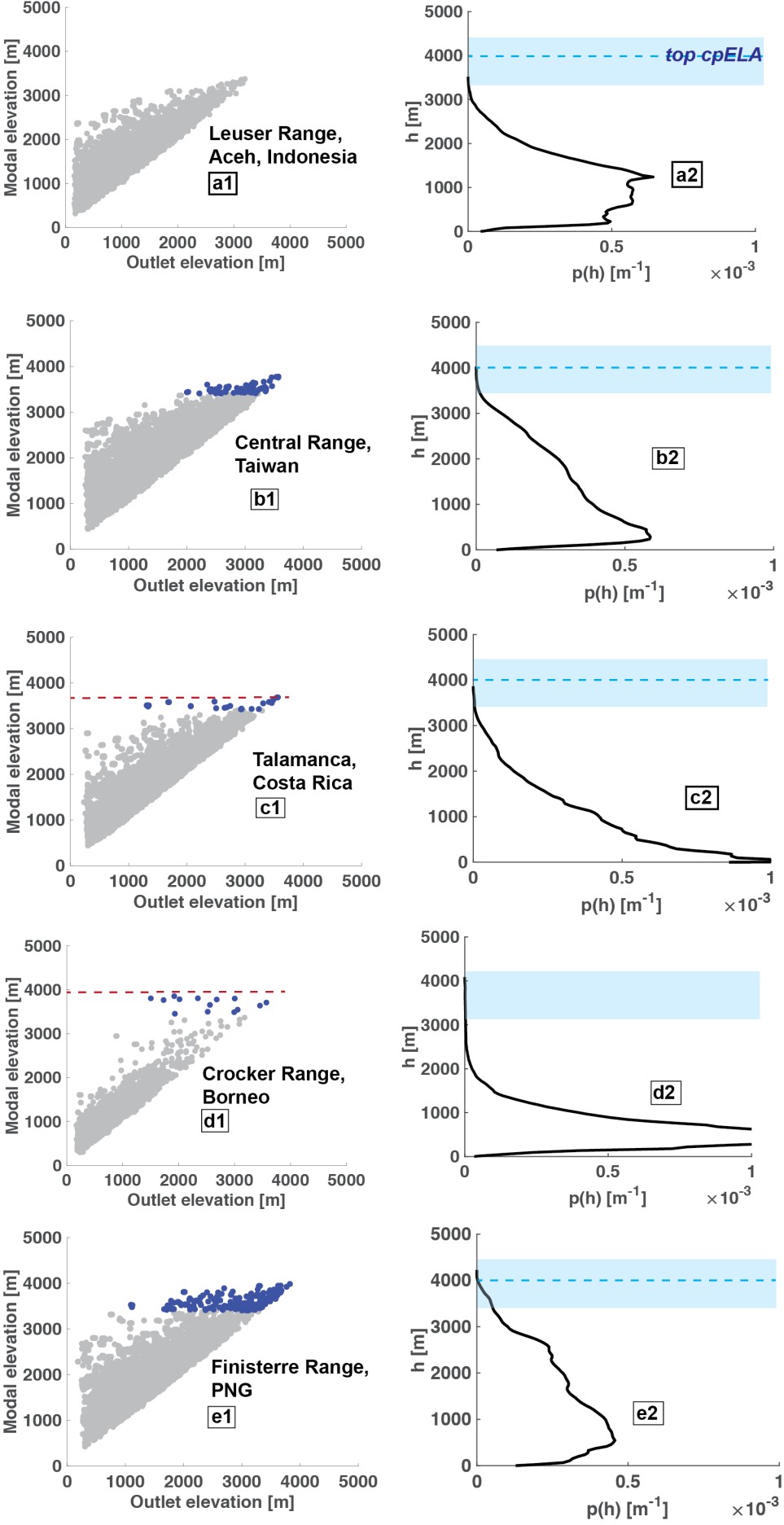

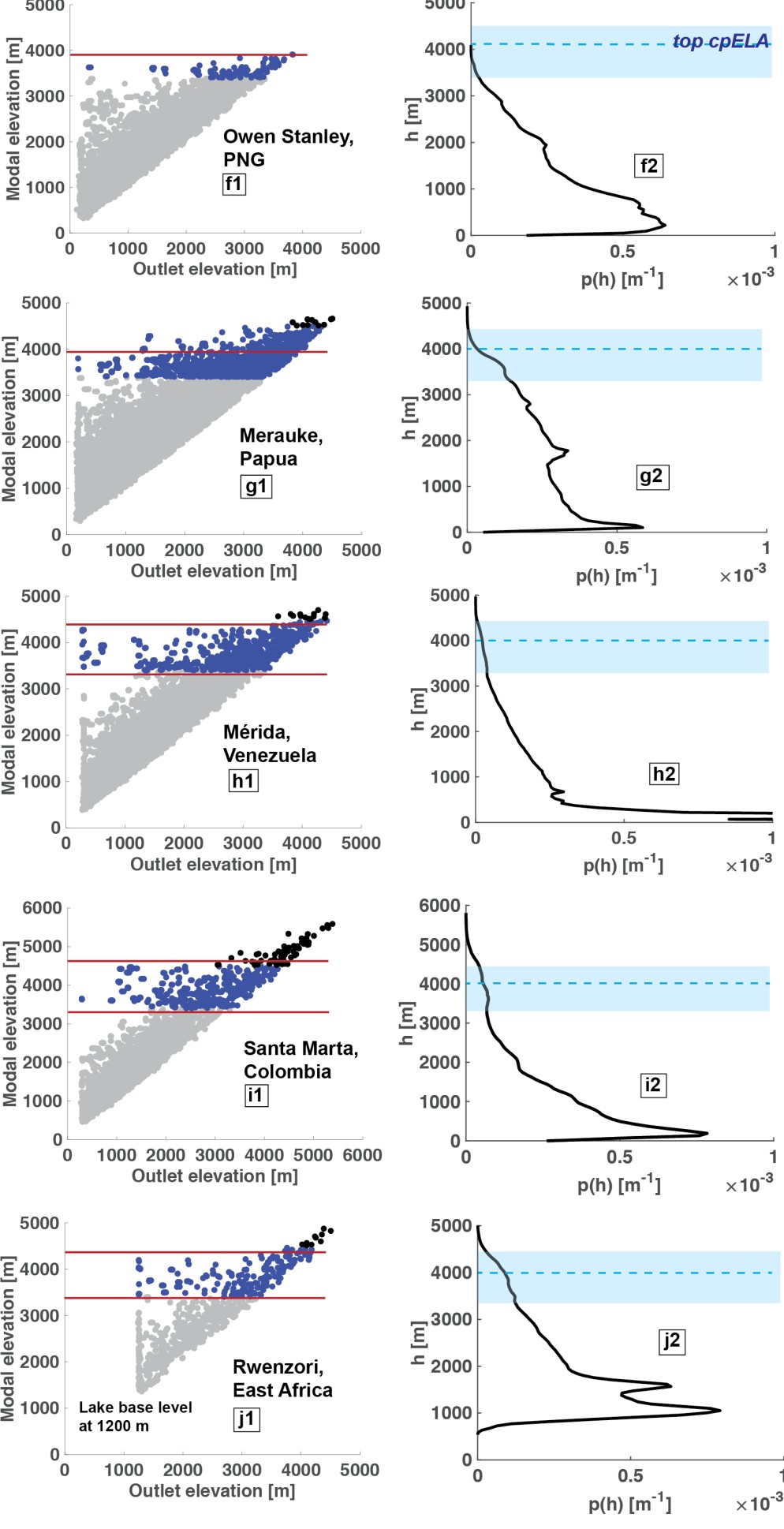

**Figure 8:** progressive hypsometry and mountain range hypsometry for selected mountain ranges (Fig. 3). **(a1-j1):** Progressive hypsometry (see Fig. 4-5) for each mountain range. Each gray point is a catchment with associated modal elevation (y-axis) and outlet (x-axis). Blue points are those catchments with a modal elevation within the range of the tropical ELA (cpELA: 3400-4000 m; wpELA 4500 m). Black points are above wpELA (4500 m). Red lines indicate left spread of modal elevation. These catchments are characterized by a strong, high modal elevation. **(a2-j2):** Mountain range hypsometry. Light blue box is range of ELA (cpELA through wpELA). Dashed blue line top of the tropical cpELA (4000 m).

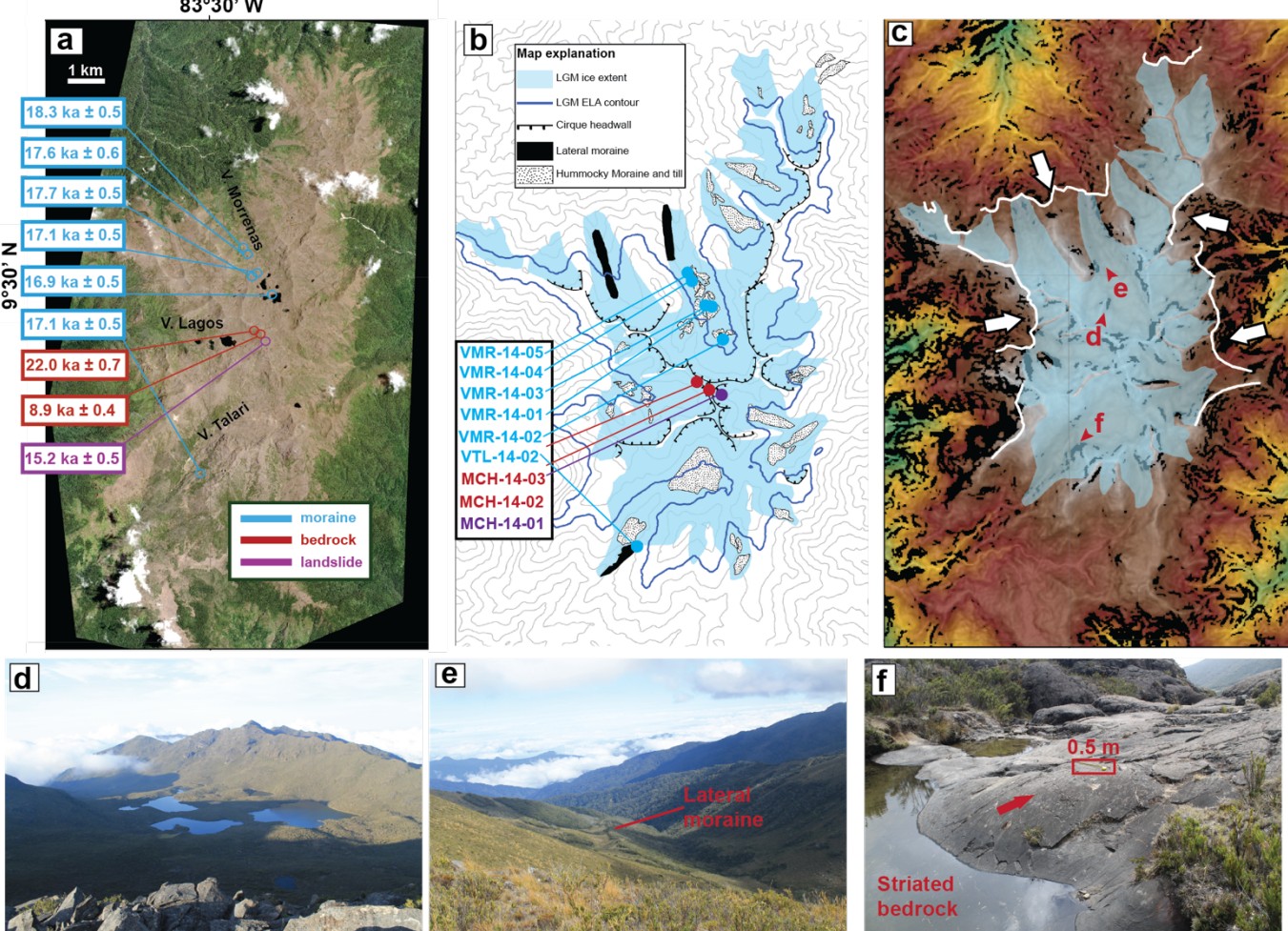

**Figure 9:** Cerro Chirripó glacial geomorphology. **(a)** Worldview-2 image with [10]Be exposure age dates. **(b)** Glacial geomorphic map of Chirripó. Heavy blue line is contour of estimated LGM ELA (3500 m). Sample locations are same is in (a). Contour interval is 100 m. Sample names correspond with field photos in Fig. 7. **(c)** SRTM DEM (green through red). Steep slopes (>30°) in black. Mapped ice extent is light blue. White lines are escarpments, and white arrows indicate direction of scarp encroachment. Red letters and arrows indicate vantage orientation of field photos (d-f). **(d)** View of cirque floor in Valle de las Morrenas. Lakes are dammed by recessional moraines. **(e)** 2 km-long lateral moraine at the base of Valle de las Morrenas. **(f)** Striated bedrock in Valle Talari. Red arrow indicates direction of ice flow.

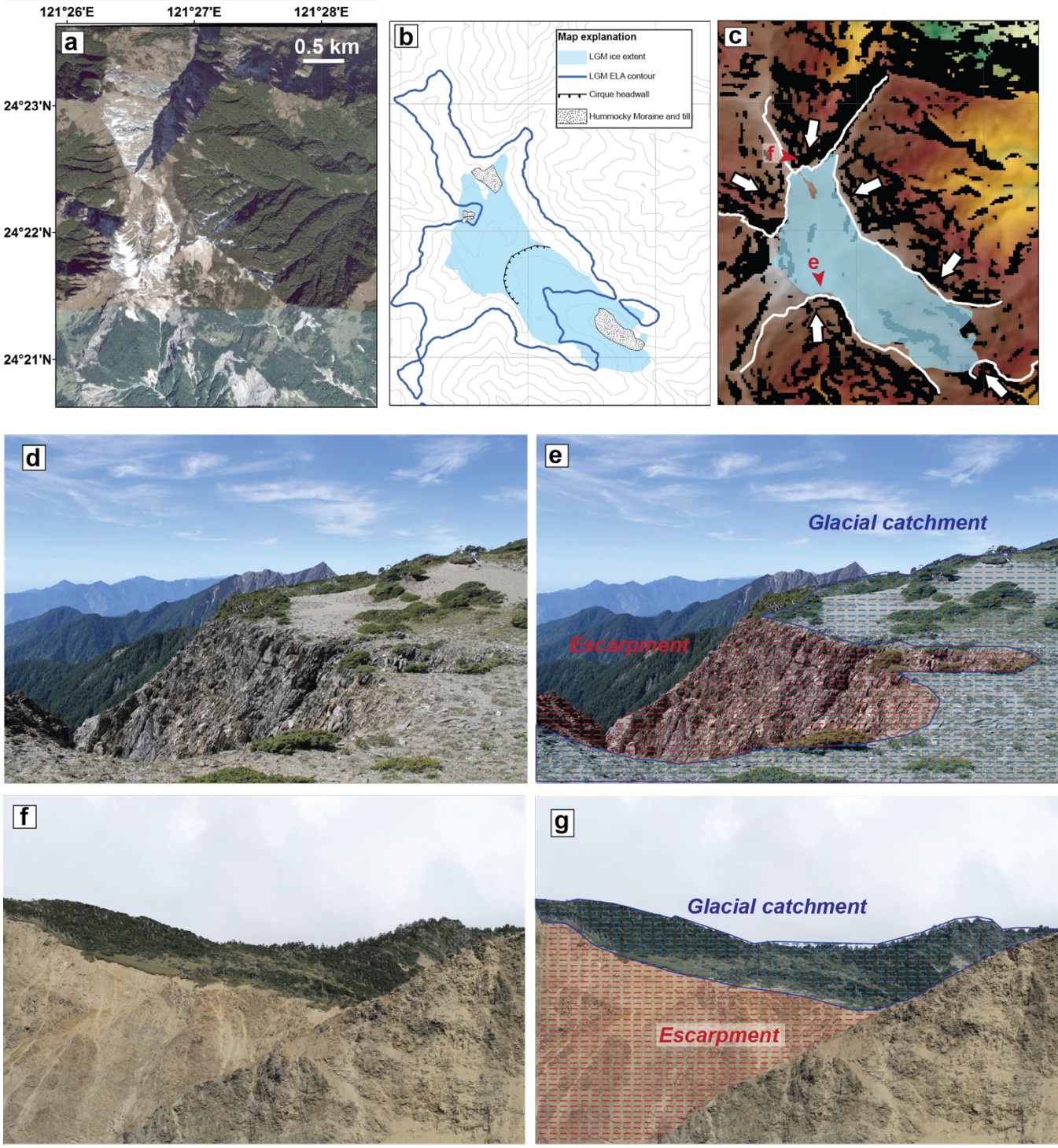

**Figure 10:** Nanhudashan glacial geomorphology. **(a)** Satellite image of Nanhudashan (Google Earth). **(b)** Glacial geomorphic map of Nanhudashan. Heavy blue line is contour of estimated LGM ELA at 3400 m. Contour interval is 100 m. **(c)** SRTM DEM (green through red). Light blue is mapped ice extent. White lines are escarpments and white arrows indicate direction of scarp encroachment. **(d-g)** Field photos of mapped escarpments. Blue and red zones in **e** and **g** correspond to glaciated zone and escarpment. Vantage point for photo **d/f** labeled with red arrow in **c**.

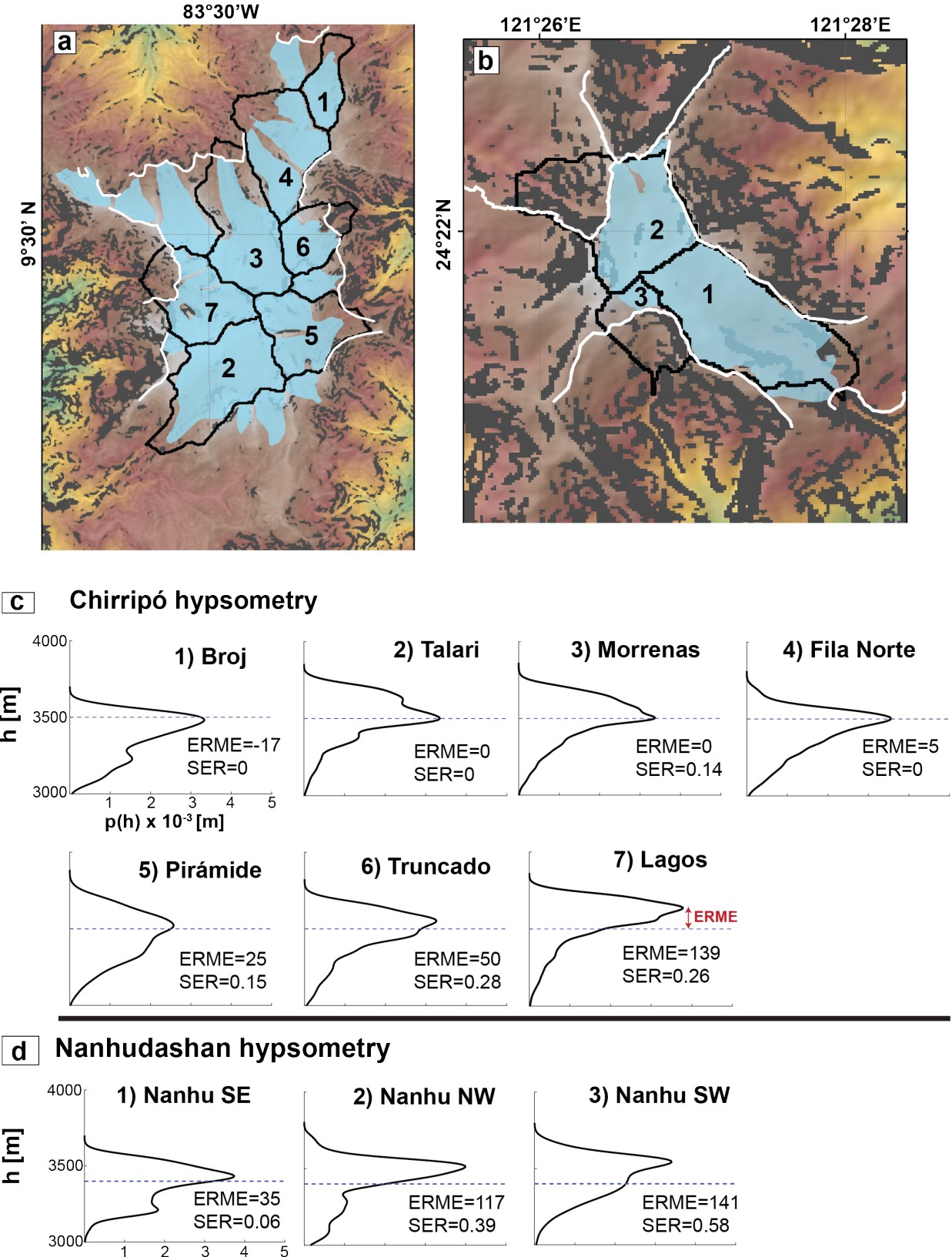

**Figure 11:** Glacial valley hypsometry at Chirripó and Nanhudashan. **(a-b)** DEM (green through red) of Cerro Chirripó and Nanhudashan with LGM ice extent (blue) and post-glacial scarp (white). Steep slopes (>30°) in black. Numbered valleys correspond with black catchment outlines. **(c-d)** Hypsometry for catchments labeled in (a-b).

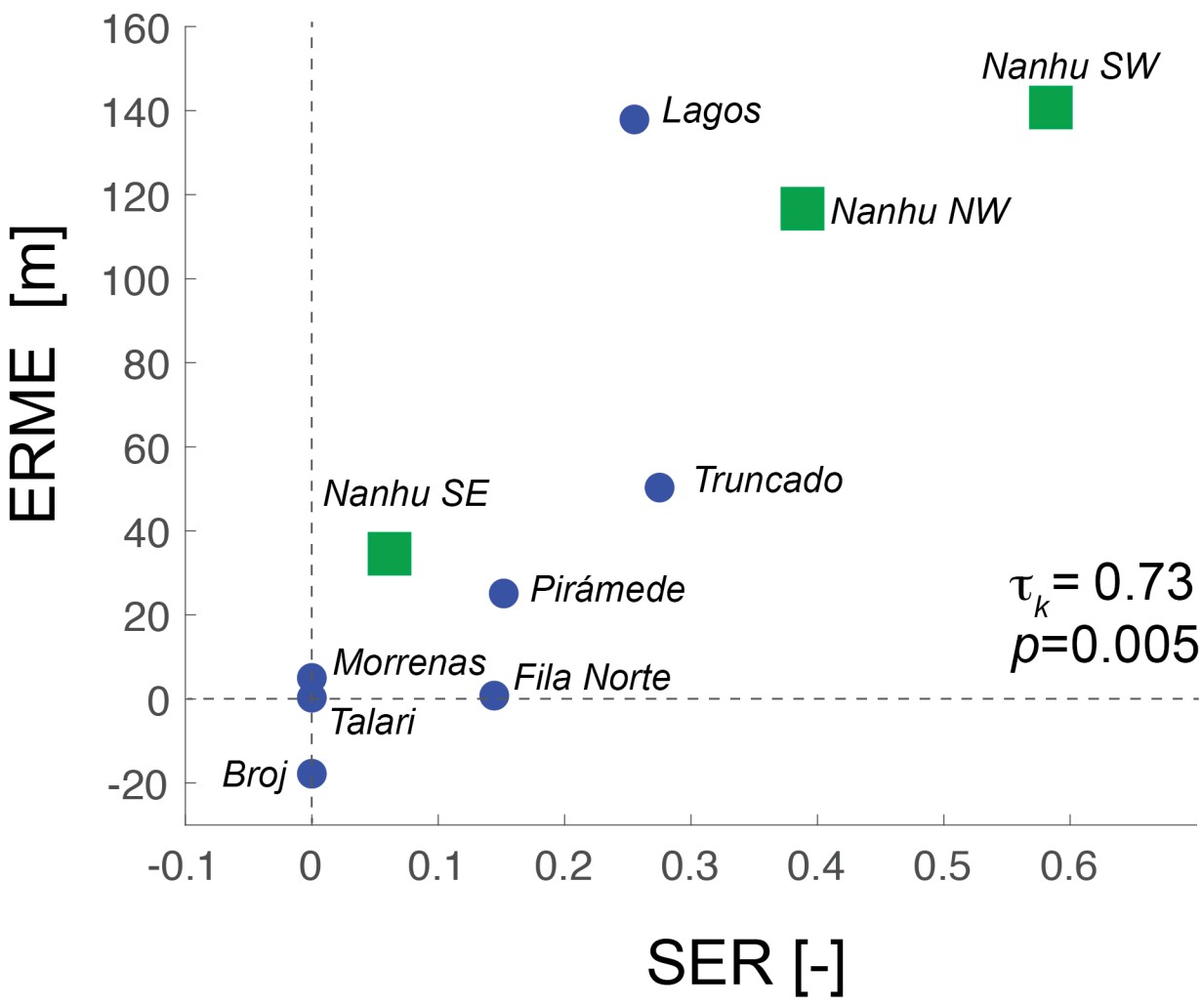

**Figure 12:** ELA-Relative Modal Elevation (ERME) and Scarp Encroachment Ratio (SER) for each catchment at Chirripó (blue) and Nanhudashan (green) (names found in Fig. 11). Kendall's tau (0.73) reported for entire data set. One-tailed significance test yields a p-value of 0.005.

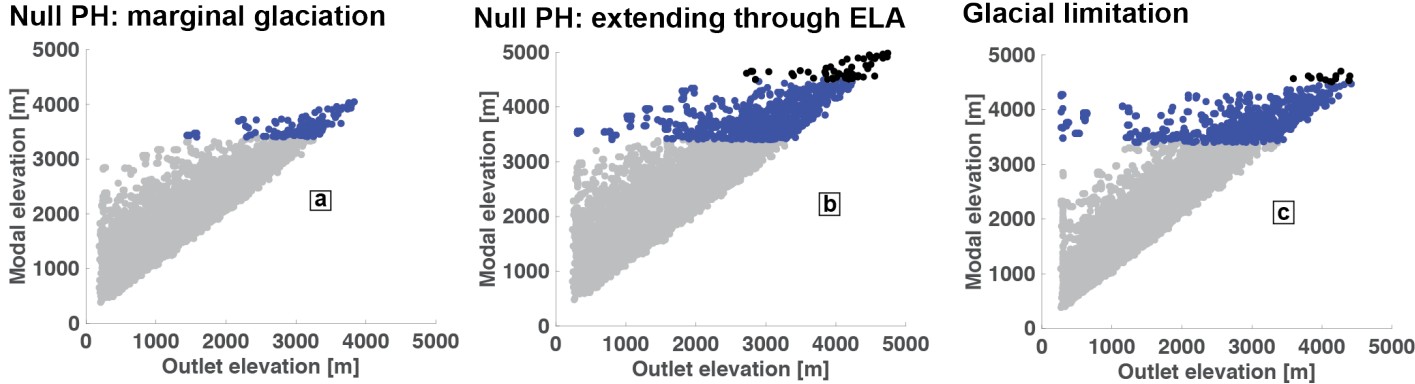

**Figure 13:** Progressive hypsometry for mountains uplifted through ELA and for glacial limitation. **(a)** Null model in which glacial erosion is either absent or negligible, or if present has been largely destroyed by interglacial processes (to the extent that it is not recorded in the PH) **(b)** As rock mass continues to rise through the ELA, glacial erosion is still unable to leave a lasting PH record, either due to its limited efficacy or because of effective scarp encroachment. **(c)** Glacial limitation: continued uplift through the ELA results in the growth of landscapes at the ELA and the establishment of a permanent glacial base-level. Glacial landscapes are preserved enough to leave a strong signal of glacial limitation in PH.

| Name | ID | Left-spread (LS) | LS-ELA match | sub-ELA Gap | Glacial features |
|---|---|---|---|---|---|
| Leuser Range | a | No | No | No | – |
| Central Range | b | No | No | No | z,y,w |
| Talamanca | c | Yes | No | No | z, x, w, u |
| Crocker Range | d | Yes | No | No | z, u |
| Finisterre Range | e | No | No | No | z,x,w,u |
| Owen Stanley | f | Yes | Yes | No | z,y,x,w,u |
| Merauke | g | Yes | No | No | z,y,x,w,u |
| Mérida | h | Yes | Yes | Yes | z,y,x,w,u |
| Santa Marta | i | Yes | Yes | Yes | z,y,x,w,u |
| Rwenzori | j | Yes | Yes | Yes | z,y,x,w,u |

**Table 1**: Summary of evidence for glacial limitation. **Left-spread (LS)** refers to high elevation hypsometric maxima with low-elevation outlets in progressive hypsometry (Fig. 8a1-j1; Fig. 13c). **LS-ELA match:** The strength of the match between the left spread and the ELA is assessed in Fig. 8a1-j1. **sub-ELA Gap:** The sub-ELA Gap refers to the absence of left spread below the ELA. **Glacial features:** z: *cirques*; y: *terminal moraines*; x: *lateral moraines*; w: *retreat moraines*; u: *overdeepenings*. Glacial features in each place have been documented in the literature and were confirmed by the authors in the available imagery (e.g., Google Earth).

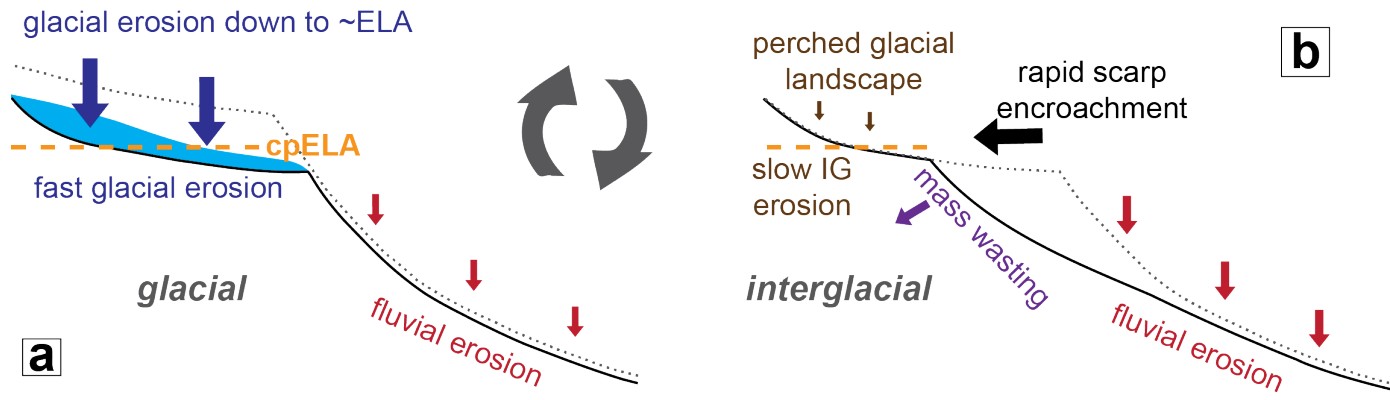

**Figure 14:** Schematic of glacio-fluvial limitation. **(a)** ELA acts as a perched base-level. Glacial erosion expands terrain near the ELA. Glacial erosion slows to zero in the ablation zone, and blocks fluvial incision. Below glacier terminus, fluvial incision continues. **(b)** Escarpments below glaciated landscapes drive headward and remove glaciated terrain.