# Peer review of "Glacial limitation of tropical mountain height"

_Earth Surface Dynamics, 2018_

## Referee Comment (RC1) · P.A. van der Beek (Referee) · 14 Jun 2018

Cunningham et al. report morphometric analyses of high-elevation mountainous massifs in Costa Rica (Cerro Chirripo) and Taiwan (Nanhudashan) to argue that these show hypsometric maxima at the elevation of the ELA during glacial advances, a characteristic of peak erosion by the "glacial buzzsaw". They also report 9 new cosmogenic 10Be ages from Cerro Chirripo to show that glacial moraines and sculpted bedrock are roughly contemporaneous with the LGM.

This is a controversial topic. Numerous authors have enthusiastically adopted the "glacial buzzsaw" concept but others have provided critical assessments of some of the observations put forward to support it. The authors provide a fairly balanced rep-

resentation of the argument in their introduction. Nevertheless, pushing the idea to encompass tropical mountains is a bold step. It appears to me that this manuscript has been in the reviewing circuit for a while now and I believe it deserves to be published, if only to have the idea out and open to critical discussion. However, I feel that the authors could, and should, back up their arguments with much better and more detailed documentation.

For the Cerro Chirripo, the problem is that the morphometric observations alone cannot discriminate between the "glacial buzzsaw" interpretation and the more conservative interpretation (put forward by Morell et al., 2012) that the high-elevation low-relief landscape represents a "relict" landscape, preserved and passively uplifted since the onset of Cocos Ridge subduction ∼3 My ago. The authors argue that the coincidence in elevation, both between the two studied examples in Costa Rica and Taiwan and with the elevation of the glacial-maximum ELA, supports the glacial buzz-saw interpretation. However, there is no way for the reader to assess this argument, as the ELA elevation for Costa Rica is only cited from (partly grey) literature, and is not given at all for Taiwan. We would like to see a detailed geomorphic map for the Cerro Chirripo, showing the elevations of the different glacial features discussed, as well as field photos showing some of these features. There are some in the Supplementary Information (and actually some more convincing ones on the first author's website blog) but these should be part of the main paper. A glacial-maximum ELA estimate of 3500 m seems on the low end for a site at <10 °N; for instance, glacial ELA estimates for the Mérida Andes in Venezuela, at approximately the same latitude, vary between ∼3600-4000 m (Stansell et al., 2007; although some estimates on the wet SE side of that mountain range descend to <3500 m). So again, more discussion and justification of these numbers seems important. A similar discussion is required for Nanhudashan; this site is at 24°N in a different geographic and climatic setting, so why should we expect a similar ELA elevation? Note also that final glacial retreat in Nanhutashan appears to have occurred much later, in the Holocene (Carcaillet et al., 2007; Siame et al., 2009).

Likewise, it is not very clear what was sampled for cosmogenic isotope analysis and why. Showing the sample sites on a geomorphic map would help significantly, as would moving some of the field photos from the Supplementary material to the main text.

In the model proposed by the authors, glacial "buzz-cutting" during cold periods competes with scarp encroachment during interglacial times (as illustrated in cartoon style in fig. 6). However, it is not clear what would drive continued scarp encroachment in this model? As the fluvial landscape below the knickpoints has a typical concave form, any lowering of the glacial landscape during "buzz-cutting" would tend to lower the slopes below the knickpoints, which does not favour scarp retreat. The authors argue for "outward spreading" of the perched glacial landscapes but, in the absence of significant deposition, it is not clear how that would work. This appears like a weak point in their argument, as these knickpoints are more directly explained in the "remnant landscape" model. One could envisage the authors' model in case of continuous rapid uplift and fluvial downcutting, which is the case in Taiwan (I do not know the Costa Rica case sufficiently well to comment on this). But in this case, the scarp retreat would be independent of the glacial "buzz-cutting" and would happen anyway (which it does; pretty much every hill slope in the Taiwan Central Range is affected by landsliding). In that case, the glacially affected high-elevation low-relief parts of the landscape are just transients that are rapidly erased and one can question their significance for overall long-term landscape development. This part of the model clearly requires some more elaboration.

More specific comments, tied to page/line number:

p. 2 / l. 15 (and elsewhere): some of the wording in the manuscript ("we add a new spin to the story ...") makes it sound like the objective here is to "push" a "nice story" instead of seeking truth, which is what science is (should be) about. This is probably not the authors' intention and the writing is simply a bit too colloquial in places, but you should really try to avoid such phrasing.

p. 3/ l. 20-24. The authors should be aware of a recent re-analysis (Schildgen et al., in press) that has shown the Herman et al. (2013) results to be flawed by a "spatial correlation bias", in which spatial variations in exhumation rates are translated into temporal increases by their model. Therefore, the thermochronometric record can no longer be used as support for increased erosion rates during Quaternary glaciations. Also, note that the Shuster et al. (2011) study argued for rapid glacial-valley incision (i.e. analogous to what Valla et al. (2011) argued for in the Western Alps) and does not pertain to glacial "buzz-cutting".

p. 5 / l. 24: "narrative" – see comment on p. 2/l. 15 above.

p. 6/ l. 1-7: this needs to be backed up by field photos and a geomorphic map.

p. 7 / l. 1-5: similarly, a map of the Nanhudashan area showing the occurrence of these glacial forms would be useful.

p. 7 / l. 20: Shuster et al. (2011) focused on glacial valley incision, not on cirque retreat.

p. 8 / l. 27: "our conclusions are not affected by the choice of production rate or scaling"; without any justification, this is a rather empty statement. I would suggest to either delete it or to provide supporting data.

p. 9 / l. 10-12: a slope map would help to demonstrate and justify the location of these erosional scarps.

p. 10 / l. 18-20: can you elaborate on what this statement is based on?

p. 11 / l. 12: "unrealistically" appears as a strange word choice for assessing data. What you probably mean is that this age, which is significantly younger than the LGM, implies that the surface must have been buried. Nothing unrealistic about that . . .

p. 12 / l. 3: the glacial ELA elevation in Taiwan has not been demonstrated or even discussed at this point.

p. 12 / l. 25-26: this statement requires justification.

p. 14 / l. 6: what do you mean by "tile-scale"?

p. 14 / l. 10-12: I don't think this statement has been demonstrated. One could just as easily argue, even within the context of this model, that the mountain belt elevation hovers around an elevation that is set by the relative efficiency of tectonic uplift versus (glacial or fluvial) erosion – it is lowered a bit during glacial times and uplifted during the transient post-glacial period of scarp encroachment.

p. 14 / l. 24-25: this is a fairly bold statement that extrapolates the findings and interpretations from Nanhudashan to all of the Taiwan Central Range. To do this, you would at a minimum need to show that the rest of the Central Range is equally affected by glacial erosion of the highest peaks and shows similar morphometry. In my understanding, glacial features in Taiwan have only been described from Nanhudashan.

p. 14 / l. 31-32: how would the glacial "buzz-cutting" "prime" the landscape for rapid horizontal scarp encroachment? See general comment above.

Fig. 1: it would be nice to have an uncluttered DEM image with an elevation scale (as well as a horizontal scale and indications of latitude and longitude). The glacial extent and the location of the scarps could be moved to the satellite image of fig. 1a (or better, could be part of a geomorphological map). The inset location map is close to unreadable.

References (other than those cited in the manuscript): Carcaillet, J., L. L. Siame, H. T. Chu, D. L. Bourlès, W. C. Lu, J. Angelier, and P. Dussouillez (2007), First cosmic ray exposure dating (in situ produced 10Be) of the late pleistocene and holocene glaciation in the Nanhutashan Mountains (Taiwan), Terra Nova, 19(5), 331–336, doi:10.1111/j.1365-3121.2007.00756.x. Schildgen, T.F., P.A. van der Beek, H.D. Sinclair, and R.C. Thieder (2018), Spatial correlation bias in late-Cenozoic erosion histories derived from thermochronology, Nature, in press. Stansell, N. D., P. J. Polissar, and M. B. Abbott

(2007), Last glacial maximum equilibrium-line altitude and paleo-temperature reconstructions for the Cordillera de Mérida, Venezuelan Andes, Quat. Res., 67(1), 115–127, doi:10.1016/j.yqres.2006.07.005.

14/06/2018 Peter van der Beek

---

## Editor Comment (EC1) · D. L. Egholm (Editor) · 30 Jul 2018

As associate editor, I would first like to thank you (authors) for sending the manuscript to ESurf and Peter van der Beek for providing a thorough review. It has unfortunately not been possible for me to secure additional reviews, but the review by Peter van der Beek provides a number of relevant points and constructive ideas. I encourage you to use all the reviewer comments to revise the manuscript including adding better and more detailed documentation to support the hypotheses presented.

In addition to the reviewer comments I list below some additional reflections of my own:

General comments:

Most previous studies of mountain range height and glacial erosion have used correlations between ELA and max topography/hypsometric maxima along climatic gradients caused by temperature or precipitation to infer that glacial erosion influences mountain range height. To me such spatial correlations provide a stronger argument than the two isolated cases presented here. We know from global compilations of topography and ELA that many exceptions to the overall trend exist for numerous reasons. I therefore encourage you to expand your study and collect data from more tropical ranges. Do any of the tropical ranges stand high above the ELA? Or do the two cases documented here indeed represent a general pattern? That two selected ranges have heights that match the estimated ELA can easily be a coincidence. Even worse: Were the ranges selected for this study because they happen to have heights that match the ELA? You need to show us more data to answer such questions and to support the general points made.

Regarding the topographical analysis you compute the hypsometry for individual catchments (focused hypsometric analysis) instead of simply computing the hypsometry of a large area (the full range, or anything above a certain elevation). While this may open for more detailed insights, it also has disadvantages when it comes to hypsometric maxima, because a catchment defined by flow routing should always have a hypsometric maxima somewhere in between the max and min elevations in the catchment. Hypsometry of a catchment may therefore differ from the hypsometry of a mountain range, which can have a hypsometric max close to baselevel. Your use of catchments at different scales only partly address this issue, and to me mountain range hypsometry is just a simpler metric to understand and use. Alternatively, you could also compare with focused hypsometries of catchments where there are no signs of glacial erosion. Do they have the same type of maximum or are they notably different? It would be useful to also see longitudinal profiles of valleys with and without evidence of glacial erosion.

I recommend that you also address the height of the ridges above the ELA. The ridges on the plateau are rather low and I would expect them to be higher, if glacial erosion

around LGM was the main erosion mechanism at high elevation. Pedersen et al. (Geomorphology 122, p. 129-139, 2010) showed how ridge height above ELA seemingly depends on the rate of tectonic uplift. Tectonic uplift rates are high in both these ranges, so what keeps the ridges down to few hundred meters above the estimated ELA? Could it be periglacial slope processes, and would they have enough time to operate in the Holocene?

More specific questions:

Page 3 Line 29: I do not see how it can be a provocative statement that glacial erosion limits the height of mountains – erosion does that. Please rephrase to explain the provocative part.

Page 4 line 5-10: This paragraph unfortunately repeats a misunderstanding that I think started with Hall & Kleman (2014): The glacial buzzsaw mechanism does not rely on horizontal erosion, and I do not think that any of the computational landscape models that you cite (e.g. Anderson, 2006; Egholm et al., 2009; MacGregor et al., 2009) even have horizontal erosion. The link between ELA and hypsometry arises because (vertical) glacial erosion is downwards limited by the mass balance of the glaciers (Egholm et al., 2009). Small glaciers do not erode deeper than the ELA because they cannot exist there. Larger glaciers can, however, because the ice flux into them keeps them alive well below the ELA. That larger glaciers cut deeper and faster than cirque glaciers is therefore not surprising, and not at all in conflict with models for the glacial buzzsaw. These two elements of a glacial landscape go hand in hand.

Page 6, line 10: It would be good to have an uncertainty estimate for the ELA. It is important here because the differences in hypsometric maxima are rather small.

Page 7, line 21: Why not record the aggregate of many valleys? Sounds good to me.

Page 9, line 31: This is where the uncertainty on the ELA becomes relevant.

Page 11, line 4: I do not think that you are constraining the timing of glacial erosion

here. Your (few) boulder samples may constrain timing of deglaciation, but the (even fewer) bedrock samples do not show any clear pattern.

Page 14, line 30 and many other places including the title: Why not just write "erosion" instead of "buzzcutting"? I don't think we really need more "buzzwords" than we already have.

Please consider the review comments carefully and submit point-by-point responses if you are willing to and interested in revising the manuscript.

Best regards David L Egholm Associate editor

---

## Author Comment (AC1) · 14 Sep 2018

**Response to Reviewer #1**

We thank Reviewer 1 (Peter van der Beek) for his thorough review, specifically for the helpful suggestions for additional figures, for highlighting weak points in our discussion, and for drawing our attention to some important references that were omitted. The suggested figures will eminently improve the clarity and quality of the paper, as will an updated discussion about general evidence for (and controversy surrounding) the existence of the "glacial buzzsaw."

*Cunningham et al. report morphometric analyses of high-elevation mountainous massifs in Costa Rica (Cerro Chirripo) and Taiwan (Nanhudashan) to argue that these show hypsometric maxima at the elevation of the ELA during glacial advances, a characteristic of peak erosion by the "glacial buzzsaw". They also report 9 new cosmogenic 10Be ages from Cerro Chirripo to show that glacial moraines and sculpted bedrock are roughly contemporaneous with the LGM.*

*This is a controversial topic. Numerous authors have enthusiastically adopted the "glacial buzzsaw" concept but others have provided critical assessments of some of the observations put forward to support it. The authors provide a fairly balanced representation of the argument in their introduction. Nevertheless, pushing the idea to encompass tropical mountains is a bold step. It appears to me that this manuscript has been in the reviewing circuit for a while now and I believe it deserves to be published, if only to have the idea out and open to critical discussion. However, I feel that the authors could, and should, back up their arguments with much better and more detailed documentation.*

*For the Cerro Chirripo, the problem is that the morphometric observations alone cannot discriminate between the "glacial buzzsaw" interpretation and the more conservative interpretation (put forward by Morell et al., 2012) that the high-elevation low-relief landscape represents a "relict" landscape, preserved and passively uplifted since the onset of Cocos Ridge subduction ~3 My ago.*

The submitted manuscript focuses on the Talamanca Range of Costa Rica, where we identify low-relief glacial landscapes above 3000 m and centered around an estimated ELA of 3500 m. In a zone between 3000 m and 3500 m there is often a pronounced slope break that separates steep fluvial landscapes from lower-sloping glacial landscapes.

Morell et al. (2012) identified knickpoints at ~2200 m ± 300 m, far below the glacial landscapes in Costa Rica. The presence of these knickpoints and other lines of evidence have supported the argument that a rapid increase in the rate of crustal deformation starting after 3 Ma drove the uplift of a landscape of 1-1.5 km of relief, with the lower slopes of this relict landscape preserved as a low-relief surface at around 2200 m. Morell et al. (2012) explicitly refer to low-relief topography above ~2000 m but *below* 3000 m as evidence of this uplifted landscape, with "high, isolated peaks above 3000 m." The glacial landscapes we describe are all above 3000 m, and cannot be linked to the inferred base of the relict landscape at ~2200 m. The low-relief topography we observe near the ELA in Costa Rica is thus not presently explained by "more conservative interpretation."

In our early unpublished (but public) work (e.g., conference abstracts, blogs) we posed glacial erosion as an alternative to the uplift of a relict landscape as an explanation for the origin of high

elevation, low-relief topography, but we now recognize that these are not mutually exclusive mechanisms. The discussion has evolved substantially since then, and in the submitted manuscript we are careful not to suggest that the glacial landscapes are related to the base of the apparent uplifted, relict landscape between 2000 and 3000 m.

*The authors argue that the coincidence in elevation, both between the two studied examples in Costa Rica and Taiwan and with the elevation of the glacial-maximum ELA, supports the glacial buzzsaw interpretation. However, there is no way for the reader to assess this argument, as the ELA elevation for Costa Rica is only cited from (partly grey) literature, and is not given at all for Taiwan.*

LGM ELA estimates produced from geomorphic mapping and ice surface reconstruction from around the tropics are generally between 3400-3800 m (e.g., Hastenrath, 2009; Mark et al., 2005) and this range of variability is consistent on a wide geographic scale. To demonstrate the widespread consistency, we list a small number of mountain ranges with paleo-ELA estimates within the cited range: New Guinea highlands (Prentice et al., 2005), eastern Africa (Kaser and Osmaston, 2002; Kelly et al., 2014) and the northern Andes (Stansell et al., 2007). There are some tropical ranges with a paleo-ELA estimate quite above this span of elevations, such as the Cordillera Blanca of Peru (4300 m, Smith et al., 2008), but in such places there are usually local circumstances that promote a higher ELA, such as a relatively dry climate associated with strong precipitation gradient in the Peruvian Andes.

The climate of Costa Rica and Taiwan is reasonably similar to other places from around the tropics whose LGM ELA estimate is between 3400-3800 m. Even before a detailed analysis of regional ELA estimates in Costa Rica and Taiwan, it is safe to assume that their LGM ELA was within this range.

We agree that the inferred position of the ELA in both landscapes must be presented in more detail, including visually in figures. In addition, more discussion is needed to demonstrate our level of confidence in the estimates provided. However, we emphasize that the LGM ELA estimates provided in the submitted manuscript are well within the expected range of tropical LGM ELA variability.

The LGM ELA of 3500 m cited from the literature for Costa Rica is, in our opinion, trustworthy. This estimate was replicated by two independent groups, who published their findings in *Quaternary Research* (Orvis and Horn, 2000) and *GSA Bulletin* (Lachniet and Seltzer, 2002). We do cite some examples from grey literature, so as to be thorough and give credit to the long history of research at Cerro Chirripó. The older work we cite is certainly worth considering, given that, e.g., Weyl (1955) and Hastenrath (1973) were serious workers, even though they lacked various luxuries afforded to modern scientists.

The later work, including Orvis and Horn (2000) and Lachniet and Seltzer (2002) is thorough glacial geomorphology, and uses standard techniques to estimate the ELA from preserved glacial landforms. Our work added surface exposure age dates to some of the landforms, and we found that they are broadly LGM.

The estimated for the ELA in Taiwan is given on p.7/l.5; however, more information is clearly needed.

ELA estimation in Taiwan is admittedly more difficult than in Costa Rica for multiple reasons. As we argue in Sect. 5.3, scarp encroachment into glaciated landscapes in Taiwan is far more severe than in Costa Rica, and we propose that for this reason the maximum extent of glaciation in Taiwan is difficult to ascertain. Previous work has indicated the presence of a glacial diamict at 2250 m in a valley flanking Nanhudashan, far below unambiguous glacial valleys (Hebenstreit and Böse, 2006). Landforms above this elevation are preserved better.

Another problem is that the timing of glaciation in Taiwan is ambiguous. The most unambiguous dated landforms are somewhat younger than global LGM (e.g., Hebenstreit et al., 2011; Siame et al., 2012). Other ages from a neighboring glacial valley in Hsuehshan (20 km west of Nanhudashan) are much older than the global LGM (Cui et al., 2002).

Finally, high rates of rock uplift complicate the estimation of the paleo-ELA in Taiwan (and elsewhere, for that matter, but most severely in Taiwan) since ELA estimation is based on reconstructions of preserved glacial remnants. We address this complexity with regard to our proposed model in the submitted manuscript.

Using the present configuration of glaciated valleys, Hebenstreit (2006) estimated an ELA of 3355 at Nanhudashan, specifically employing the terminal to summit altitudinal method (TSAM). Hebenstreit (2006) also used TSAM to estimate an undated ELA of 3400 m at Yushan, a third glaciated massif in southwest Taiwan. Other work has used the maximum vertical extent of lateral moraines in both Hsuehshan (Cui et al., 2002) and at Yushan (Böse, 2004) to propose an ELA of ~3400 m. The age associated with these ELA estimates remains difficult to pin down, and may be more closely associated with the Late Glacial than with the global LGM.

An appropriate LGM ELA estimate for Taiwan is therefore certainly below 3500 m. A reasonable low estimate might be 3200 m. Hebenstreit (2006) suggested the lowest possible limit of the LGM ELA to be 2775 m, but this is certainly an extreme lower limit.

To address this comment, we will add:
1) more detailed geomorphic maps of both focus sites in Costa Rica and Taiwan;
2) discussion on the procedures for estimating the ELA in Taiwan;
3) discussion of other ELA estimates from around the tropics.

*We would like to see a detailed geomorphic map for the Cerro Chirripo, showing the elevations of the different glacial features discussed, as well as field photos showing some of these features. There are some in the Supplementary Information (and actually some more convincing ones on the first author's website blog) but these should be part of the main paper.*

This is an excellent suggestion, and we agree that such figures would greatly improve the quality and clarity of the manuscript. We will introduce these figures in the revised manuscript.

*A glacial-maximum ELA estimate of 3500 m seems on the low end for a site at <10˚N; for instance, glacial ELA estimates for the Mérida Andes in Venezuela, at approximately the same latitude, vary between ∼3600-4000 m (Stansell et al., 2007; although some estimates on the wet SE side of that mountain range descend to <3500 m). So again, more discussion and justification of these numbers seems important. A similar discussion is required for Nanhudashan; this site is at 24°N in a different geographic and climatic setting, so why should we expect a similar ELA elevation?*

To address these concerns more discussion of the ELA is clearly needed, and we will provide this in the revised manuscript.

There are climatic differences between Costa Rica and Taiwan, but these are probably not as extreme as might be indicated by their difference in latitude. Costa Rica has weak thermal seasons (diurnal air temperature fluctuation exceeds the annual variability) and the mean annual air temperature at high elevations (3475 m) is 7.6°C (Lachniet and Seltzer, 2002). Thermal seasonality in Taiwan is greater, and monthly annual air temperature at Nanhudashan today ranges from -2.6°C in winter to 8.2°C in summer (Klose, 2006).

The question of why the LGM ELA in Costa Rica and Taiwan was similar is somewhat beyond the scope of this paper, as we simply rely on geomorphic evidence that indicates its position in both places. However, if paleo-glaciers in Taiwan responded mostly to summer temperature, and glaciers in Costa Rica were more strongly controlled by annual mean temperature, as is common in the deep tropics (e.g., Kaser and Osmaston, 2002), then the temperature that is most important for determining the position of the ELA is similar in both places today (8.2°C in Taiwan, 7.6°C in Costa Rica).

There is substantially more complexity in the pattern of the position of the LGM ELA in Mérida Range than is plausible in Costa Rica or Taiwan. Stansell et al. (2007) argue that a combination of local variability in precipitation, cloud cover, and aspect drove a change in the LGM ELA of ~600 m over the 25 km width of the massif. The width of glacial landscapes in the massifs we present (~5 km) are a small fraction of this width, and we propose that there is no important change in the position of the ELA at either Cerro Chirripó or at Nanhudashan. That said, the range of ELA proposed by Stansell et al. (2007) for the Mérida Range encompasses the ELA estimates we present for Costa Rica and Taiwan.

*Likewise, it is not very clear what was sampled for cosmogenic isotope analysis and why. Showing the sample sites on a geomorphic map would help significantly, as would moving some of the field photos from the Supplementary material to the main text.*

We agree. We will move these field photos to the main text in the revised manuscript.

*In the model proposed by the authors, glacial "buzz-cutting" during cold periods competes with scarp encroachment during interglacial times (as illustrated in cartoon style in fig. 6). However, it is not clear what would drive continued scarp encroachment in this model? As the fluvial landscape below the knickpoints has a typical concave form, any lowering of the glacial landscape during "buzz-cutting" would tend to lower the slopes below the knickpoints, which does not favour scarp retreat. The authors argue for "outward spreading" of the perched glacial landscapes but,*

*in the absence of significant deposition, it is not clear how that would work. This appears like a weak point in their argument, as these knickpoints are more directly explained in the "remnant landscape" model.*

The scarp encroachment we propose arises from differential erosion between glacial landscapes and the fluvial landscapes flanking them. The ELA is an effective base level for glaciated landscapes, and glacial incision and headward erosion creates low-sloping valleys near this elevation. Low-slopes in the vicinity of the ELA promote slow erosion during ice-free periods—an effect that is not felt in the flanking fluvial network. We propose that the slope break between glacial and fluvial landscapes grows as erosion in the flanking fluvial network drives on during warm periods, and eventually grows into the escarpments we observe.

"Outward spreading" is poor phrasing on our part, and we rewrite this sentence in the revised manuscript. This phrasing was intended to reference to the concept of cirque backcutting, detailed, for example, in Oskin and Burbank (2005). We envision cirque glaciers eroding headward, leaving behind relatively low-sloping topography near the ELA, and generating erosion fronts (knickzones) *in situ* below the glacial limit.

Finally, the knickpoints that mark the break between glacial and fluvial landscapes around 3000 m are not the same knickpoints as those thought to represent the break between the uplifted relict landscape and steeper topography below it at ~2200 m.

*One could envisage the authors' model in case of continuous rapid uplift and fluvial downcutting, which is the case in Taiwan (I do not know the Costa Rica case sufficiently well to comment on this). But in this case, the scarp retreat would be independent of the glacial "buzz-cutting" and would happen anyway (which it does; pretty much every hill slope in the Taiwan Central Range is affected by landsliding). In that case, the glacially affected high-elevation low-relief parts of the landscape are just transients that are rapidly erased and one can question their significance for overall long-term landscape development. This part of the model clearly requires some more elaboration.*

Glacial landscapes in these mountain ranges do appear to be transient features, and their "significance in the overall, long-term landscape development" should be questioned, although it never has been before. Our central point has been that if glacial landscapes are transient features, then glaciation has either happened once, at the LGM, in isolated patches in both ranges, and never anywhere else or at any prior time in either range. The alternative we propose is that glaciation has happened there repeatedly.

From this and other comments, it is now clear to us that more discussion is needed on possible landscape evolution scenarios that could produce the topographic patterns we observe today, which we categorize into three broad groups:

1) **LGM glacial erosion alone**
   a. Both Taiwan and Costa Rica have been rising throughout the Pleistocene and during the last 20 kyr have reached elevations necessary for glaciation
   b. Steady state has not been achieved yet in either place, and *true* peak elevations have yet to be attained (i.e., both ranges will reach elevation >>cold-phase ELA)
   c. Glaciation has happened only once in both places, and has been spatially limited to Chirripó in Costa Rica and Nanhudashan, Yushan, and Hsuehshan in Taiwan

2) **Glacial decoration**
   a. Fluvial incision has kept pace with rock uplift and has reached topographic and flux steady state
   b. This elevation happens to put enough rock mass above the cold-phase ELA for periodic glaciation
   c. Glacial erosion happens in short bursts but is insufficient to lower the landscape below the fluvially-achieved steady state
   d. Glacial erosion also does little to affect the fluvial erosion below the glacial limit, and thus has virtually no effect on the steady state elevation, and the ambient erosion rate is largely invariant between glaciated and non-glaciated catchments

3) **Glacio-fluvial limitation (GFL)**
   a. During the Pleistocene rock mass has repeatedly crossed the cold-phase ELA
   b. Glacial erosion occurs once sufficient rock mass pushes through the ELA
   c. Periodic glaciation has been sufficient to reduce catchments above the ELA down to the ELA
   d. Glacially eroded catchments are disconnected from flanking fluvial network
   e. Fluvial escarpments propagate headward, remove glacial landscapes

Again, we have introduced Scenario 3 in the submitted manuscript, and we acknowledge that Scenario 1 and Scenario 2 are also possible. We recommend that the likelihood of Scenario 3 would increase if similar patterns were to be observed in more ranges than those presented. To this end, we extend our analysis to more mountains ranges in the revised manuscript (the details of this approach are outlined in our response to David Egholm).

*More specific comments, tied to page/line number:*

*p. 2 / l. 15 (and elsewhere): some of the wording in the manuscript ("we add a new spin to the story . . .") makes it sound like the objective here is to "push" a "nice story" instead of seeking truth, which is what science is (should be) about. This is probably not the authors' intention and the writing is simply a bit too colloquial in places, but you should really try to avoid such phrasing.*

Our objective is certainly not to push a nice story. We rephrase colloquial statements in the revised manuscript.

*p. 3/ l. 20-24. The authors should be aware of a recent re-analysis (Schildgen et al., in press) that has shown the Herman et al. (2013) results to be flawed by a "spatial correlation bias", in which spatial variations in exhumation rates are translated into temporal increases by their model.*

*Therefore, the thermochronometric record can no longer be used as support for increased erosion rates during Quaternary glaciations. Also, note that the Shuster et al. (2011) study argued for rapid glacial-valley incision (i.e. analogous to what Valla et al. (2011) argued for in the Western Alps) and does not pertain to glacial "buzz-cutting".*

Thank you for pointing us to this reference. We will rewrite this section of the introduction.

*p. 5 / l. 24: "narrative" – see comment on p. 2/l. 15 above.*

*p. 6/ l. 1-7: this needs to be backed up by field photos and a geomorphic map.*

We agree. These will be included in a revised manuscript.

*p. 7 / l. 1-5: similarly, a map of the Nanhudashan area showing the occurrence of these glacial forms would be useful.*

We agree. We include a geomorphic map of glacial features at Nanhudashan in the revised manuscript.

*p. 7 / l. 20: Shuster et al. (2011) focused on glacial valley incision, not on cirque retreat.*

Thank you for pointing this out. We will rewrite this sentence.

*p. 8 / l. 27: "our conclusions are not affected by the choice of production rate or scaling"; without any justification, this is a rather empty statement. I would suggest to either delete it or to provide supporting data.*

The supporting data were included in the supplementary file, and simply show that different scaling regimes alter calculated exposure ages by <2 kyr. Since our central conclusion is that glacial valleys at Chirripó were subject to LGM erosion, we conclude this relatively small range of variability does not affect our overall conclusion.

*p. 9 / l. 10-12: a slope map would help to demonstrate and justify the location of these erosional scarps.*

We will separate the slope map from the DEM in Fig. 1, and add a new figure of a stand-alone slope map.

*p. 10 / l. 18-20: can you elaborate on what this statement is based on?*

We sought a way to quantify the effect of scarp encroachment into glacial catchments, specifically, what segment of glaciated valleys have been affected by scarp encroachment. To do so, we required some reference point for individual catchment outlet elevations, which we chose to be 3000 m. We chose this elevation on the basis that the lowest moraines observed extend to about 3000 m elevation.

*p. 11 / l. 12: "unrealistically" appears as a strange word choice for assessing data. What you probably mean is that this age, which is significantly younger than the LGM, implies that the surface must have been buried. Nothing unrealistic about that . . .*

Fair point. We will reword in a revised manuscript.

*p. 12 / l. 3: the glacial ELA elevation in Taiwan has not been demonstrated or even discussed at this point.*

Discussed above.

*p. 12 / l. 25-26: this statement requires justification.*

Cerro Chirripó is the highest peak in Costa Rica, and Nanhudashan is the highest peak in N.E. Taiwan. We looked at the catchments that share these peaks and found that they have a hypsometric maximum at the LGM ELA, a tell-tale sign of significant glacial erosion.

*p. 14 / l. 6: what do you mean by "tile-scale"?*

1°x1° SRTM DEM tiles. Egholm et al. (2009) used these tiles in their global analysis.

*p. 14 / l. 10-12: I don't think this statement has been demonstrated. One could just as easily argue, even within the context of this model, that the mountain belt elevation hovers around an elevation that is set by the relative efficiency of tectonic uplift versus (glacial or fluvial) erosion – it is lowered a bit during glacial times and uplifted during the transient post-glacial period of scarp encroachment.*

See comment about three landscape evolution scenarios above.

*p. 14 / l. 24-25: this is a fairly bold statement that extrapolates the findings and interpretations from Nanhudashan to all of the Taiwan Central Range. To do this, you would at a minimum need to show that the rest of the Central Range is equally affected by glacial erosion of the highest peaks and shows similar morphometry. In my understanding, glacial features in Taiwan have only been described from Nanhudashan.*

To clarify, LGM glacial features in Taiwan are best preserved at Nanhudashan, and for this reason we focused our analysis there. More ambiguous LGM glacial features have also been reported at Hsuehshan (Cui et al., 2002) and Yushan (e.g., Hebenstreit et al., 2011).

The peaks of all of massifs in Taiwan are within ~500 m of the estimated ELA, and on the scale of individual valleys there is very little area above the ELA at all. We have proposed that an explanation for the relatively constant elevation of peaks throughout Taiwan—many of them close to the ELA—can be explained by a glacial erosion acting at different, isolated peaks throughout the Pleistocene. Wherever glacial erosion does occur, a low-sloping, transient glacial landscape is left behind and eventually wiped from the landscape by fluvially-driven scarp propagation.

We respectfully disagree with the statement that "at a minimum" we would need to show that "the rest of the Central Range is equally affected by glacial erosion." In our model, glacial erosion stops the highest parts of the landscape at the ELA, and glacial landscapes are then "reworked" into the flanking fluvial network by scarp encroachment, reducing their preservation potential. To infer glacio-fluvial height limitation in Taiwan we rely on similar evidence from comparable mountain ranges (in the submitted manuscript only the Talamanca Range, but as we detail in our response to David Egholm, we will add substantially to the discussion of evidence of glacio-fluvial height limitation throughout the tropics). This statement is thus not premised on our observations from Taiwan alone.

*p. 14 / l. 31-32: how would the glacial "buzz-cutting" "prime" the landscape for rapid horizontal scarp encroachment? See general comment above.*

See answer above with regard to scarp encroachment.

*Fig. 1: it would be nice to have an uncluttered DEM image with an elevation scale (as well as a horizontal scale and indications of latitude and longitude). The glacial extent and the location of the scarps could be moved to the satellite image of fig. 1a (or better, could be part of a geomorphological map). The inset location map is close to unreadable.*

We agree. We will include updated figures in a revised manuscript.

**References**

Böse, M.: Traces of glaciation in the high mountains of Taiwan. *Developments in Quaternary Science*, *2*(PART C), 347–352, https://doi.org/10.1016/S1571-0866(04)80141-6, 2004.

Hastenrath, S.: On the Pleistocene glaciation of the Cordillera de Talamanca, Costa Rica, Zeitschrift für Gletscherkunde and Glazialgeologie, 9, 105–121, 1973.

Hastenrath, S.: Past glaciation in the tropics: Quaternary Science Reviews, 28, 790–798, doi:10.1016/j.quascirev.2008.12.004, 2009.

Hebenstreit, R.: Present and former equilibrium line altitudes in the Taiwanese high mountain range: Quaternary International, 147, 70–75, doi:10.1016/j.quaint.2005.09.008, 2006.

Hebenstreit, R., Böse, M., and Murray, A.: Late Pleistocene and early Holocene glaciations in Taiwanese mountains: Quaternary Science Reviews, 147, 76-88, doi:10.1016/j.quaint.2005.09.009, 2006.

Hebenstreit, R., Ivy-Ochs, S., Kubik, P.W., Schlüchter, C., and Böse, M.: Late glacial and early Holocene surface exposure ages of glacial boulders in the Taiwanese high mountain range: Quaternary Science Reviews, 30, 298–311, doi:10.1016/j.quascirev.2010.11.002, 2011.

Kaser, G., and Osmaston, H.: *Tropical glaciers*, Cambridge University Press, 2002.

Kelly, M.A., Russel, J.M., Baber, M.B., Howley, J.A., Loomis, S.E., Zimmerman, S., Nakileza, B., Lukaye, J.: Expanded glaciers during a dry and cold Last Glacial Maximum in equatorial East Africa: Geology, 42, 519-522, doi:10.1130/GS35421.1, 2014.

Klose, C.: Climate and geomorphology in the uppermost geomorphic belts of the Central Mountain Range, Taiwan: Quaternary International, 147, 89–102, doi:10.1016/j.quaint.2005.09.010, 2006.

Lachniet, M. S., and Seltzer, G.O.: Late Quaternary glaciation of Costa Rica: GSA Bulletin, 114, 547 558, doi.org/10.1130/0016-7606(2002)114<0547:LQGOCR>2.0.CO;2, 2002.

Mark, B.G., Harrison, S.P., Spessa, A., New, M., Evans, D.J.A., and Helmnes, K.F.: Tropical snowline changes at the last glacial maximum: A global assessment: Quaternary International, 138-139, 168-201, doi:10.1016/j.quaint.2005.02.012, 2005.

Morell, K. D., Kirby, E., Fisher, D.M., and van Soest, M.: Geomorphic and exhumational response of the Central American Volcanic Arc to Cocos Ridge subduction: Journal of Geophysical Research, 117, B04409, doi:10.1029/2011JB008969, 2012.

Orvis, K. H., and Horn, S.P.: Quaternary glaciers and climate on Cerro Chirripó, Costa Rica, Quaternary Research, 54, 24–37, doi:10.1006/qres.2000.2142, 2000.

Oskin, M., and Burbank, D.W.: Alpine landscape evolution dominated by cirque retreat, Geology, 33, 933–936, doi:10.1130/G21957.1, 2005.

Prentice, M.L., Hope, G.S., Maryunani, K., and Peterson, J.A.: An evaluation of snowline data across New Guinea during the last major glaciation, and area-based glacier snowlines in the Mt. Jaya region of Papua, Indonesia, during the Last Glacial Maximum: Quaternary International, 138-139, 93-117, doi:10.1016/j.quaint.2005.02.008, 2005.

Smith, J.A., Mark, B.G., Rodbell, R.T.: The timing and magnitude of mountain glaciation in the tropical Andes: Journal of Quaternary Science, 23, 609–634, doi: 10.1002/jqs.1224, 2008.

Stansell, N. D., Polissar, P. J., and Abbott, M. B., Last glacial maximum equilibrium-line altitude and paleo-temperature reconstructions for the Cordillera de Mérida, Venezuelan Andes: Quaternary Research, 67, 115–127. doi:10.1016/j.yqres.2006.07.005, 2007.

Weyl, R.: Vestigios de una glaciación del Pleistoceno en la Cordillera de Talamanca, Costa Rica: Instituto Geografico de Costa Rica, Informe Trimestral, 9–32, 1955.

---

## Author Comment (AC2) · 14 Sep 2018

**Response to Reviewer #2**

*As associate editor, I would first like to thank you (authors) for sending the manuscript to ESurf and Peter van der Beek for providing a thorough review. It has unfortunately not been possible for me to secure additional reviews, but the review by Peter van der Beek provides a number of relevant points and constructive ideas. I encourage you to use all the reviewer comments to revise the manuscript including adding better and more detailed documentation to support the hypotheses presented.*

We appreciate the very thorough and detailed reviews provided by both and you and Peter van der Beek, and for steering our manuscript through a very helpful review process. We have considered all comments carefully, and propose a set of revisions to address the ideas and suggestions provided by you and Peter van der Beek.

*In addition to the reviewer comments I list below some additional reflections of my own:*

*General comments:*
*Most previous studies of mountain range height and glacial erosion have used correlations between ELA and max topography/hypsometric maxima along climatic gradients caused by temperature or precipitation to infer that glacial erosion influences mountain range height. To me such spatial correlations provide a stronger argument than the two isolated cases presented here. We know from global compilations of topography and ELA that many exceptions to the overall trend exist for numerous reasons. I therefore encourage you to expand your study and collect data from more tropical ranges. Do any of the tropical ranges stand high above the ELA? Or do the two cases documented here indeed represent a general pattern? That two selected ranges have heights that match the estimated ELA can easily be a coincidence. Even worse: Were the ranges selected for this study because they happen to have heights that match the ELA? You need to show us more data to answer such questions and to support the general points made.*

To be clear, the decision to focus on the Talamanca Range (CR) and the Central Range (TW) did not begin with their ELA-height match. Rather, we were initially struck by the following: even though global scale observations (e.g., Egholm et al., 2009, Fig. 1C) of the ELA-height match include the tropics, glacial erosion has *not been proposed as a mechanism for limiting tropical mountain height*. Our goal was to explore this possible mechanism through the study of tropical landscapes that are potentially the most prone to glacial limitation. We deduced that high mountain ranges (peak elevations above 2000 m) that are tectonically active, rapidly eroding, and circumferentially well-connected to external base-level forcing are likely the best recorders of glacial erosion. Those that best match such criteria are:

  1) Finisterre Range, Papua New Guinea
  2) Owen Stanley Range, Papua New Guinea
  3) Merauke Range, Papua
  4) Central Papua New Guinea Highlands
  5) Crocker Range, Borneo
  6) Leuser Range, Aceh Province of Indonesia
  7) Central Range, Taiwan

8) Sierras Madre, Mexico/Guatemala
9) Talamanca Range, Costa Rica/Panama
10) Santa Marta, Venezuela
11) Mérida Range, Venezuela

There is a tradeoff between the geographic scale and the degree of detail in any analysis. In the submitted manuscript, we choose to apply detailed analysis of the Talamanca Range in Costa Rica (CR) and the Central Range of Taiwan (TW) because they exemplify the selection criteria, and because there are good constraints on the timing of deglaciation in both ranges. We recognize the need to apply a wider geographic analysis, and to this end we discuss the full list of tropical ranges above before describing in detail the rationale for focusing on these two in the revision.

Furthermore, in the revision we present a three-step analysis, progressively thinning the targeted mountain ranges and introducing more detailed analysis at each step. This three-step approach is detailed below:

**Step 1: Analyze the hypsometry of 1°x1° SRTM tiles of entire range**

In this step, we adapt the approach of Egholm et al. (2009), who found the hypsometric maximum (specifically, the highest modal elevation of multi-modal elevation distributions) of every glaciated tile between 60°N and 60°S. The tropical ranges we list above were included in this analysis, and the majority do not have a hypsometric maximum near the lower limit of late-Pleistocene ELA fluctuation. However, the hypsometries of many 1°x1° tiles covering these ranges do show evidence of *truncation* at or very close to the ELA. This nuance is obscured when only the hypsometric maximum of the tile is recorded.

In Fig. 1 and Fig. 2, below, we plot two examples (Merauke Range, Papua and Mérida Range, Venezuela) of tile hypsometry mosaics that are provided in the revised manuscript for each of the 11 mountain ranges listed above. Note that in both example mountain ranges plotted below there are peaks far above the LGM ELA (such as the presently glaciated Puncak Jaya in the Merauke Range of Papua).

Our goal is to use tile hypsometry to identify a subset of mountain ranges that may be subject to glacial limitation or, as we propose for CR and TW in the submitted manuscript, glacio-fluvial limitation, even if their tile hypsometry does not reveal a broad aerial extent of glacial landscapes.

**Step 2: Analyze hypsometry of selected mountain range on a progressively smaller scale.**
In this analysis, we will choose a subset of ~5 ranges in which we analyze hypsometry over a range of scales. The details of this method are presented in response to a comment below (Figs. 3-6).

**Step 3: Focused analysis of glacial landscapes in CR and TW.**
We will conclude our analysis with a detailed look at glacial landscapes in the Talamanca and Taiwan in order to assess the interplay between glacial and fluvial erosion in tropical highlands and the possibility of glacio-fluvial limitation of mountain height in the tropics.

[Figure]

**Figure 1: Tile hypsometry of the Merauke Range.** A: Merauke range DEM. 0-3500 m is light blue through red, 3500-4500 m is dark blue to white. B-G: Elevation pdf of 1°x1° tiles, labeled by southwest corner of tile. Dashed red lines are approximate bounds of ELA variability in New Guinea (e.g., Prentice et al., 2005). Puncak Jaya (4844 m) is located in tile 5S, 137E.

[Figure]

**Figure 2: Tile hypsometry of the Mérida Range.** A: Mérida range DEM. 0-3500 m is light blue through red, 3500-4500 m is dark blue to white. B-F: Elevation pdf of 1°×1° tiles, labeled by southwest corner. Dashed red lines are approximate bounds of tropical ELA variability (these bounds were estimated for the Mérida range by Stansell et al. 2007).

*Regarding the topographical analysis you compute the hypsometry for individual catchments (focused hypsometric analysis) instead of simply computing the hypsometry of a large area (the full range, or anything above a certain elevation). While this may open for more detailed insights, it also has disadvantages when it comes to hypsometric maxima, because a catchment defined by flow routing should always have a hypsometric maxima somewhere in between the max and min elevations in the catchment. Hypsometry of a catchment may therefore differ from the hypsometry of a mountain range, which can have a hypsometric max close to base-level. Your use of catchments at different scales only partly address this issue, and to me mountain range hypsometry is just a simpler metric to understand and use.*

We agree that the choice of scale in analyzing hypsometry is critically important, particularly when assessing the significance of glacial erosion in the landscape. Tile hypsometric analysis provides the benefit of comparing the aerial extent of glacial landscapes relative to flanking fluvial landscapes in a consistent domain size, but, as we have shown in the submitted manuscript, some glacial landscapes are obscured at this scale of analysis.

A central question we attempted to raise in the submitted manuscript is whether the absence of a hypsometric maximum in e.g. tile-scale analysis is indicative of the absence of (significant) glacial erosion. The tile hypsometry of CR and TW and many other tropical ranges show that glacial landscapes have a small areal extent relative to the fluvial catchments and depositional plains that surround them. Does this observation alone justify the claim that glacial erosion cannot impose the limit on mountain height in these places?

We argue that such a claim requires further justification for two reasons:

1) When the ELA is a relatively high elevation, fluvial catchments must be large (in elevation range) for glacial erosion to take place at all. Thus the absence of a hypsometric maximum at the ELA on the large scale does not indicate the absence of glacial erosion, or even glacial limitation (or glacio-fluvial limitation).
2) Glacial erosion and fluvial erosion can act in tandem to limit mountain height to the ELA, with fluvially-driven escarpments attacking glacial landscapes during warm periods. This process could limit mountain height near the ELA, but would not leave sufficient terrain at and above the ELA for a hypsometric maximum to manifest at this elevation.

We thus are left with the following problem: if mountain ranges truncate at elevations near the (cold-phase) ELA, as indicated in tile-scale hypsometric analysis for places like CR and TW, what scale of analysis is appropriate to assess the potential role of glacial erosion in limiting mountain height? We argue that different scales of analysis are needed to assess the overall significance of glacial landscapes in environments like those found in the tropics.

In step 2 of the updated analysis, we introduce a modified method of focused hypsometric analysis, which we call "progressive hypsometry." The method involves a progressive measurement of hypsometry along nested catchments whose outlets span from the lowest to the highest elevations in a mountain range. We present the method as a way to assess the significance of hypsometric maxima found across a mountain range.

*Progressive Hypsometry:*

1) **Segment landscape into large catchments that link the main divide to base level.**
2) **Map channel network:**
   a. map drainage using D8, steepest descent flow routing (Schwanghart and Scherler, 2014)
   b. define a channel network using an arbitrary flow accumulation area threshold $A\_c$
   c. traverse downstream from each channel head $i=1...N$ to the catchment exit to define a set of $N$ along-channel pixel chains
   d. extend each chain $i$ upstream from its channel head to the drainage divide by following path of greatest flow accumulation area, ensuring that each pixel chain spans the full range of elevation from ridge to exit
3) **Map progressive hypsometry (PH) along this network:**
   a. traverse each chain $i$ upstream from the exit (shared by all chains)
   b. map along each chain a nested series of subcatchments, one per channel pixel *j(i)*
   c. calculate hypsometry for each nested subcatchment, and record its modal elevation *h_mode_j* and its outlet elevation *h_out_j*
   d. record as a set of $i=1...N$ sequences of *[h_out_j(i),h_mode_j(i)]* pairs
4) **Identify all PH "benches", characteristic nested-catchment modal elevations:**
   a. perform change-point detection along each chain $i=1...N$ to locate and define large jumps in *h_mode* at each *h_out*
   b. define the outlet elevation *h_out* at each jump as *h_change*
   c. designate the groups of between-jump modal elevations *{h_mode}* as "benches"
   d. define each bench modal elevation *h_bench = min{h_mode}*
5) **Identify the principal PH benches, their locations and jump heights:**
   a. concatenate all $N$ sequences of *[h_change_k(i),h_bench_k(i)]*
   b. record as a single array of $M$ jump-bench pairs *[h_change_m,h_bench_m]* where *m=1...M*

Fig. 3 demonstrates progressive hypsometry steps 1-4; Fig. 4 demonstrates the final progressive hypsometry step 5.

We propose this method as a solution to the problem of scale in hypsometric analysis: particularly when the aim is to assess the importance of glacial erosion. Rather than choosing one scale, either large or small, and checking for a hypsometric maximum at the ELA, we find the hypsometric maximum of catchments at virtually all scales in a targeted mountain range.

The proposed method does not fully address the criticism that glacial landscapes are not aerially extensive, because the hypsometric maximum of a catchment in many cases is a narrow elevation band, and not indicative of any unusual process at that elevation. Rather, progressive hypsometry identifies parts of the landscape where there is reasonable suspicion that glacial erosion has taken place by finding catchments with a hypsometric maximum at the ELA. More detailed analysis of such catchments is required to confirm that glacial erosion has taken place there. For places like Cerro Chirripó and Nanhudashan, the manuscript as written describes such analysis—although as this and Peter van der Beek's reviews have made clear, more clear documentation of these landscapes is needed.

[Figure]

**Figure 3: Example of progressive hypsometry along one chain.** A: DEM of Cerro Chirripó, 0-4000 m is blue through white. Light blue is glacial extent at Chirripó. Dark blue streamline is example of one chain along which progressive hypsometry is performed. B: Modal elevation (hypsometric maximum) for each progressively higher outlet elevation along dark blue streamline. C-F: Elevation pdf of catchments associated with jump in modal elevation (hypsometric maximum) in B. Labels in A point to outlet elevations associated with each elevation pdf. Dashed red line is local estimated LGM ELA.

[Figure]

**Figure 4: Progressive hypsometry for large catchment.** A,E: DEM of Cerro Chirripó, CR and Nanudashan, TW, respectively. Catchments boundaries are examples of those extracted along one streamline for a single large catchment. Light blue in both plots is LGM ice extent. We chose to highlight subcatchments located at jumps modal elevation (B,F). B,F: Modal elevation (hypsometric maximum) for each progressively higher outlet elevation along dark blue streamline. C,G: All hypsometric steps from every streamline in large catchment in gray. Red points (B1-B4; F1-F6) correspond with red points on B,F.

Progressive hypsometry provides a detailed perspective on how elevation is distributed in the landscape. Below we present the tile hypsometry (Fig. 5a) and the progressive hypsometry (Fig 5b) of the central Talamanca Range, CR.

[Figure]

**Figure 5: Tile hypsometry vs. progressive hypsometry:** A: Hypsometry of Talamanca Range, SRTM tile with southwest corner 9°N, 84°W. B: Progressive hypsometry for all catchments in Talamanca SRTM tile.

In the tile hypsometry of the Talamanca it is apparent that relative to a span of elevations from sea level to ~4000 m there is very little area above ~3000 m, indicating that glacial landscapes around 3500 m occupy and very small fraction of the total mountain range, and depositional zones below 1000 m and topographic benches around 2500 m occupy the most area. Progressive hypsometry shows the distribution of elevation in *both* zones of the entire mountain range that are dominated by these prominent, low-sloping features and those that are not. For example, the progressive hypsometry plot Fig. 5b shows that some catchments have a hypsometric maximum associated with a topographic bench at 2500 m and an outlet as low as 250 m. In other catchments, sometimes with outlet elevations as low as ~1300 m, the hypsometric maximum is associated with glacial landscapes at 3500 m.

To summarize, progressive hypsometry can characterize the fine scale topographic patterns of entire mountain ranges, and can reveal features that go missed in tile hypsometry. This particular example shows the power of analyzing tile and progressive hypsometry together: on the scale of the entire Talamanca Range there is very little area above ~3000 m, but in the zone above 3000 m, the ELA (3500 m) dominates the elevation distribution.

Fig. 6 shows progressive hypsometry of three regions of the Talamanca Range, as an example of how progressive hypsometry can be deployed on a large scale. Only glaciated catchments have a hypsometric maximum near the ELA—elsewhere in the mountain range topographic benches are found at varying elevations, but never appreciably above the ELA. Note that this figure includes primarily fluvial catchments, and that LGM glacial erosion only affected a small part of the mountain range.

We present this new method as a way to guide the assessment of glacially eroded landscapes in mountain ranges where glaciated valleys occupy a small fraction of the landscape. We stress that the presence of a hypsometric maximum at the ELA in a progressive hypsometry plot does not alone confirm the significance of glacial erosion or even that glacial erosion has even taken place, but rather is a reliable predictor that glacial erosion has acted in the landscape. Additional evidence is required to assess in full the quality of glacial erosion, such as the field evidence we present from Costa Rica.

The revised mansucript includes progressive hypsometry analysis of a subset of the list of 11 tropical mountain ranges provided above.

[Figure]

**Figure 6: Range-scale progressive hypsometry.** A: Talamanca Range, CR. 0-4000 m is black through white. Colored catchment boundaries are large catchments used to organize progressive hypsometry. B-D: Progressive hypsometry of corresponding to colored catchment outlines in A.

[Figure]

*Alternatively, you could also compare with focused hypsometries of catchments where there are no signs of glacial erosion. Do they have the same type of maximum or are they notably different?*

The method outline above characterizes the hypsometry of both fluvial and glacial catchments.

*It would be useful to also see longitudinal profiles of valleys with and without evidence of glacial erosion.*

We will provide these in a supplemental figure in an updated manuscript.

*I recommend that you also address the height of the ridges above the ELA. The ridges on the plateau are rather low and I would expect them to be higher, if glacial erosion around LGM was the main erosion mechanism at high elevation. Pedersen et al. (Geomorphology 122, p. 129-139, 2010) showed how ridge height above ELA seemingly depends on the rate of tectonic uplift. Tectonic uplift rates are high in both these ranges, so what keeps the ridges down to few hundred meters above the estimated ELA? Could it be periglacial slope processes, and would they have enough time to operate in the Holocene?*

Thank you for reminding us about this important reference, which is highly relevant to our work.

The height of ridges above the ELA in our focus areas are best explained by the combination of glacial erosion and scarp encroachment that act in concert to limit mountain height, and it is thus not surprising that the total relief of these glacial landscapes is relatively small. Pedersen et al. (2010) invoke a steady state balance between rock uplift and glacial erosion to explain the correlation between ridge height and uplift rate in glacially eroded landscapes at the mid-latitudes (p. 136). We argue that glacial landscapes in the tropics do not achieve a steady state balance with rates of rock uplift because they are destroyed relatively quickly by fluvial escarpments.

*More specific questions:*

*Page 3 Line 29: I do not see how it can be a provocative statement that glacial erosion limits the height of mountains – erosion does that. Please rephrase to explain the provocative part.*

Thank you for bringing this to our attention. We will rephrase in a revised manuscript.

*Page 4 line 5-10: This paragraph unfortunately repeats a misunderstanding that I think started with Hall & Kleman (2014): The glacial buzzsaw mechanism does not rely on horizontal erosion, and I do not think that any of the computational landscape models that you cite (e.g. Anderson, 2006; Egholm et al., 2009; MacGregor et al., 2009) even have horizontal erosion. The link between ELA and hypsometry arises because (vertical) glacial erosion is downwards limited by the mass balance of the glaciers (Egholm et al., 2009). Small glaciers do not erode deeper than the ELA because they cannot exist there. Larger glaciers can, however, because the ice flux into them keeps them alive well below the ELA. That larger glaciers cut deeper and faster than cirque glaciers is therefore not surprising, and not at all in conflict with models for the glacial buzzsaw. These two elements of a glacial landscape go hand in hand.*

This paragraph was in reference to Valla et al. (2011) who claimed that evidence of rapid glacial incision below the ELA "contradicted" the buzzsaw, but your comment demonstrates clearly that their findings are not necessarily such a contradiction. We will rewrite this paragraph.

In terms of "horizontal" erosion: our wording equates "horizontal" erosion with "headward" erosion, which arises from effective glacial erosion near the ELA that shifts up-valley. We view headward erosion as a form of horizontal erosion, even though none of the models cited explicitly parametrize horizontal erosion.

*Page 6, line 10: It would be good to have an uncertainty estimate for the ELA. It is important here because the differences in hypsometric maxima are rather small.*

We will include an ELA uncertainty in the revised manuscript.

*Page 7, line 21: Why not record the aggregate of many valleys? Sounds good to me.*

Hypsometric maxima at the "aggregate" scale are effectively recorded in progressive hypsometry, as large catchments are aggregates of many catchments.

*Page 9, line 31: This is where the uncertainty on the ELA becomes relevant.*

Agreed. We will address the uncertainty in a revised manuscript.

*Page 11, line 4: I do not think that you are constraining the timing of glacial erosion here. Your (few) boulder samples may constrain timing of deglaciation, but the (even fewer) bedrock samples do not show any clear pattern.*

This is a fair point. We will revise.

*Page 14, line 30 and many other places including the title: Why not just write "erosion" instead of "buzzcutting"? I don't think we really need more "buzzwords" than we already have.*

This a fair point. We will also revise this as suggested.

**References**

Egholm, D. L., Nielsen, S. B., Pedersen, V. K., and Lesemann, J.-E.: Glacial effects limiting mountain height, Nature, 460, 884–5 887, doi:10.1038/nature08263, 2009.

Prentice, M.L., Hope, G.S., Maryunani, K., and Peterson, J.A.: An evaluation of snowline data across New Guinea during the last major glaciation, and area-based glacier snowlines in the Mt. Jaya region of Papua, Indonesia, during the Last Glacial Maximum: Quaternary International, 138-139, 93-117, doi:10.1016/j.quaint.2005.02.008, 2005.

Schwanghart, W., Scherler, D.: TopoToolbox 2 – MATLAB-based software for topographic analysis and modeling in Earth surface sciences: Earth Surface Dynamics, 2, 1-7, doi: 10.5194/esurf-2-1-2014, 2014.

Stansell, N. D., Polissar, P. J., and Abbott, M. B., Last glacial maximum equilibrium-line altitude and paleo-temperature reconstructions for the Cordillera de Mérida, Venezuelan Andes: Quaternary Research, 67, 115–127. doi:10.1016/j.yqres.2006.07.005, 2007.

---

## Author Response (AR1)

Maxwell Cunningham
302B Oceanography
61 Route 9W
Palisades, NY-10964
maxwellc@ldeo.columbia.edu

David Egholm
Associate Editor, Earth Surface Dynamics

November 22, 2018

Dear Dr. Egholm,

Thank you for considering our manuscript "Glacial buzzcutting limits the height of tropical mountains." We really appreciate the detailed and engaging reviews provided by both you and Peter van der Beek, and we thank you and Earth Surface Dynamics for your guidance in revising our work.

We hereby submit the revised manuscript, now entitled "Glacial limitation of tropical mountain height", for your consideration. The critical reviews motivated us to refocus our core arguments, to augment our methods of analysis, and to expand their scope; as a result, the manuscript has undergone substantial modification. We estimate the order of the changes to the text and figures to be around 40%, including the creation of new sections and figures, the rewriting of several existing sections, and a reorganization of the paper structure; for these reasons, it was not feasible to generate a track-changed document. Instead, the changes are listed on pp. 2–3 below, followed by a detailed response to reviewers on pp. 4–15 (in which *the reviewer comments are in italics* and our responses are in plain text). We believe that by working carefully to address the comments and issues raised by the reviews we have made significant improvements to the manuscript.

The main changes are as follows. We now emphasize the concept of glacial limitation of mountain height (rather than the "glacial buzzsaw") and its relationship with a perched erosional base-level at the ELA. A major addition is the introduction of a modified form of hypsometric analysis— which we call "progressive hypsometry"—and its application in the tropics. We use this tool to evaluate the hypothesis that glacial base-levels are widespread in tropical mountains, and we tie results from this analysis to the original study of Costa Rica and Taiwan. We note that our conclusions are broadly similar to those presented in the previously submitted manuscript, but are now supported by a much broader and stronger base of data and analysis.

We hope this revision meets with your approval.

Sincerely,

Max Cunningham

**Changes to text**

*OS = old section (in original submission)*
*NS = new section (in revised submission)*

1) Title has been modified.
2) Abstract has been entirely rewritten.
3) Introduction section 1 has been expanded. Some of the text has been transferred to a new section NS2 "Evidence for glacial limitation: a review". A new subsection NS1.3 detailing the paper structure has been added.
4) OS2 "Tectonic and geomorphic setting" has been transferred to new subsections NS3.2.
5) "Methods" section OS3 has been moved to new section NS5.
6) A wholly new section NS3 has been added. This new section has a description of a study areas taken from across the tropics that are analyzed with both traditional and progressive hypsometry (the latter a methodological innovation presented for the first time in this paper).
7) A new subsection NS3.1.2 details issues regarding the ELA in the tropics.
8) OS3.2 describing surface exposure-age dating has been moved to NS5.3.2.
9) OS3.3 describing scarp mapping has been moved to NS5.3.1.
10) A short new section NS4 "Data" has been added to summarize DEM, imagery and exposure age data sources.
11) The old section OS4 "Results" has been moved to NS6.
12) The revised NS5 "Methods" includes new subsections on DEM processing (NS5.1), a complete revision of OS3.1 into NS5.2 "Hypsometry", a new subsubsection NS5.2.1 on "Range-scale hypsometry" (the traditional approach), a new subsubsection NS5.2.2 describing the new tool of progressive hypsometry, and a new subsection NS5.3 "Focus sites" which draws text from OS3.3 and OS4.1
13) The new results section NS6 includes additions covering results from range-scale (traditional) hypsometry (NS6.1) and progressive hypsometry (NS6.2), together with a reorganized description of results from studies of the two focus sites (NS6.3: Costa Rica; NS6.4: Taiwan). The comparison subsection NS6.5 is a largely unmodified OS4.3.
14) The new discussions section NS7 is a complete revision of OS5.
15) The revised conclusions section NS8 is a complete revision of OS6.

**Changes to Figures**

*OF = old figure (in original submission)*
*NF = new figure (in revised submission)*
*NT = new table (in revised submission)*

1) NF1 documenting rationale for the work has been moved from the supplement to the main text and annotated.
2) NF2 has been introduced to support expanded section 1 and NS2.
3) NF3 supports NS3, which is a description of study areas (both new and old) from across the tropics.
4) NF4 and NF5 demonstrate how "progressive hypsometry" works.
5) OF5 demonstrating scarp encroachment has been moved to NF6, and now includes field documentation of scarps.
6) NF7: field photos of $^{10}$Be sample sites, has been moved from the supplement.
7) NF8 includes entirely new results from new hypsometric analyses.
8) OF1 has been moved to NF9, and now includes a glacial geomorphic map and field photos.
9) NF10: a satellite image, glacial geomorphic map, DEM, and field photos of Nanhudashan.
10) OF2 has been moved to NF11. Minimal changes. OF3, of "hypsometry at different scales" has been removed, and is now conceptually addressed in NF8.
11) OF5 to Fig. 12: No changes.
12) NF13: a schematic explaining how to interpret the progressive hypsometry plots.
13) NT1: provides a breakdown of the evidence for glacial limitation at each of the study sites.
14) OF6 to NF14: minimal changes.

**AE Comments**

*As associate editor, I would first like to thank you (authors) for sending the manuscript to ESurf and Peter van der Beek for providing a thorough review. It has unfortunately not been possible for me to secure additional reviews, but the review by Peter van der Beek provides a number of relevant points and constructive ideas. I encourage you to use all the reviewer comments to revise the manuscript including adding better and more detailed documentation to support the hypotheses presented.*

We appreciate the very thorough and detailed reviews provided by both and you and Peter van der Beek, and for steering our manuscript through a very helpful review process. We have considered all comments carefully, and include revisions to address the ideas and suggestions provided by you and Peter van der Beek.

**Reviewer 1**

*Cunningham et al. report morphometric analyses of high-elevation mountainous massifs in Costa Rica (Cerro Chirripo) and Taiwan (Nanhudashan) to argue that these show hypsometric maxima at the elevation of the ELA during glacial advances, a characteristic of peak erosion by the "glacial buzzsaw". They also report 9 new cosmogenic 10Be ages from Cerro Chirripo to show that glacial moraines and sculpted bedrock are roughly contemporaneous with the LGM.*

*This is a controversial topic. Numerous authors have enthusiastically adopted the "glacial buzzsaw" concept but others have provided critical assessments of some of the observations put forward to support it. The authors provide a fairly balanced representation of the argument in their introduction. Nevertheless, pushing the idea to encompass tropical mountains is a bold step. It appears to me that this manuscript has been in the reviewing circuit for a while now and I believe it deserves to be published, if only to have the idea out and open to critical discussion. However, I feel that the authors could, and should, back up their arguments with much better and more detailed documentation.*

We thank Peter van der Beek for the comments, and for encouraging the publication of our work. Below we provide a detailed response to all comments.

*For the Cerro Chirripo, the problem is that the morphometric observations alone cannot discriminate between the "glacial buzzsaw" interpretation and the more conservative interpretation (put forward by Morell et al., 2012) that the high-elevation low-relief landscape represents a "relict" landscape, preserved and passively uplifted since the onset of Cocos Ridge subduction ~3 My ago.*

We address this criticism on multiple levels.

First, we articulate the concept of glacial limitation much more thoroughly (p. 5, Sec. 1.1). Our goal has been to assess whether glacial erosion limits the height of tropical mountains. In the revised manuscript we present hypsometric analysis of ten different tropical mountain ranges, including the Talamanca Range, and show that none of them have been able to push rock mass through the ELA without being subject to the introduction of a glacial base-level. An ensemble of ten mountain ranges is the central evidence that glacial erosion is effective at reducing relief above the cpELA. In any particular mountain range, the uplift of low-relief topography that is glacially eroded as it passes through the ELA does not disqualify the potential for glacial limitation.

We use the Talamanca Range of Costa Rica as a type example of a marginally glaciated mountain range, and ask: has glacial erosion been effective there? We now demonstrate more clearly that it has (Fig. 9). These and similar observations in Taiwan support our broader claim that substantial glacial erosion takes place as soon as topography reaches the cpELA, and that it continues to shape landscapes as they rises through the ELA (Fig. 8).

Second, in the Talamanca Range, we identify glacial landscapes above 3000 m and centered around an estimated ELA of 3500 m. In a zone between 3000 m and 3500 m there is often a pronounced slope break that separates steep fluvial landscapes from lower-sloping glacial landscapes. Morell et al. (2012) identified knickpoints at ~2200 m ± 300 m, far below the glacial landscapes in Costa Rica. The presence of these knickpoints and other lines of evidence have supported the argument that a rapid increase in the rate of crustal deformation starting after 3 Ma drove the uplift of a landscape of 1-1.5 km of relief, with the lower slopes of this relict landscape preserved as a low-relief surface at around 2200 m. Morell et al. (2012) explicitly refer to low-relief topography above ~2000 m but *below* 3000 m as evidence of this uplifted landscape, with "high, isolated peaks above 3000 m." The glacial landscapes we describe are all above 3000 m, and cannot be linked to the inferred base of the relict landscape at ~2200 m. The low-relief topography we observe near the ELA in Costa Rica is thus not presently explained by a "more conservative interpretation."

*The authors argue that the coincidence in elevation, both between the two studied examples in Costa Rica and Taiwan and with the elevation of the glacial-maximum ELA, supports the glacial buzzsaw interpretation. However, there is no way for the reader to assess this argument, as the ELA elevation for Costa Rica is only cited from (partly grey) literature, and is not given at all for Taiwan. We would like to see a detailed geomorphic map for the Cerro Chirripo, showing the elevations of the different glacial features discussed, as well as field photos showing some of these features. There are some in the Supplementary Information (and actually some more convincing ones on the first author's website blog) but these should be part of the main paper.*

*A glacial-maximum ELA estimate of 3500 m seems on the low end for a site at <10˚N; for instance, glacial ELA estimates for the Mérida Andes in Venezuela, at approximately the same latitude, vary between ~3600-4000 m (Stansell et al., 2007; although some estimates on the wet SE side of that mountain range descend to <3500 m). So again, more discussion and justification of these numbers seems important. A similar discussion is required for Nanhudashan; this site is at 24°N in a different geographic and climatic setting, so why should we expect a similar ELA elevation?*

We have added substantial discussion about the typical cold-phase ELA around the tropics (p. 8, Sec. 3.1.2) and cite literature that brackets the tropical cold-phase ELA to between 3400-4000 m

(this is actually the range that Stansell et al. (2007) cite for the Mérida Range). The 3400-4000 m ELA range is often observed within single, wide mountain ranges (e.g., Mérida Range, Merauke Range) but narrower mountain ranges (Talamanca Range, Finisterre Range) around the tropics always have a cold-phase ELA in this range.

We then describe the methods used to estimate the ELA at Cerro Chirripó (p. 11, l. 12-22) and throughout Taiwan (Sec. 3.2.2, p. 12, l. 5-31). We do cite work from the "grey" literature, so as to be scholarly and thorough, and to give credit to the long history of research at Cerro Chirripó and Nanhudashan, but we rely most heavily on sources published in journals such as *Quaternary Science Reviews* (Siame et al., 2007; Hebenstreit et al., 2011), *Quaternary Research* (Orvis and Horn, 2000) and *GSA Bulletin* (Lachniet and Seltzer, 2002). We also provide a glacial geomorphic map for Cerro Chirripó (Fig. 9) and for Nanhudashan (Fig. 10).

We use LGM ELA estimates for Cerro Chirripó and Nanhudashan that are presented in the literature, and it is beyond the scope of our work to assess the climatic reasons that the ELA is similar between these two places despite being separated by 13° latitude.

In summary, we have included the following elements in the revised manuscript:

1) more detailed geomorphic maps of both focus sites in Costa Rica and Taiwan;
2) discussion on the procedures for estimating the ELA in Taiwan;
3) discussion of other ELA estimates from around the tropics.

*Likewise, it is not very clear what was sampled for cosmogenic isotope analysis and why. Showing the sample sites on a geomorphic map would help significantly, as would moving some of the field photos from the Supplementary material to the main text.*

Field photos of all samples for [10]Be analysis are now presented in Fig. 7, and are labeled on the glacial geomorphic map of Cerro Chirripó on Fig. 9.

*In the model proposed by the authors, glacial "buzz-cutting" during cold periods competes with scarp encroachment during interglacial times (as illustrated in cartoon style in fig. 6). However, it is not clear what would drive continued scarp encroachment in this model? As the fluvial landscape below the knickpoints has a typical concave form, any lowering of the glacial landscape during "buzz-cutting" would tend to lower the slopes below the knickpoints, which does not favour scarp retreat. The authors argue for "outward spreading" of the perched glacial landscapes but, in the absence of significant deposition, it is not clear how that would work. This appears like a weak point in their argument, as these knickpoints are more directly explained in the "remnant landscape" model.*

We now make clear from the very beginning of the manuscript that glacial erosion introduces a perched base level near the ELA (p. 3, Sec. 1.1). Glacial erosion effectively "disconnects" fluvial landscapes in the following way: (p. 3, l. 20-24):

"Below the ELA, ice flow spreads laterally, ablates, and slows, driving sub-glacial erosion rates to zero. The (near-) ELA acts as an erosional base-level, above which ice-driven erosion pushes headward into the landscape (Fig. 2b). Glacial erosion ultimately disconnects these landscapes from fluvial base-level by blocking channel incision above the glacier terminus—an elevation where glacial erosion is also least effective."

The scarp encroachment we propose arises from the disconnection of glacial landscapes from fluvial landscapes. As we describe throughout the manuscript (p.3-4, Sec. 1.2; p. 20, l.11-18) this disconnection happens at high elevations in the tropics because of the high ELA. Whatever lowering takes place by glacial erosion is thus not sufficient to weaken the effect of scarp encroachment.

"Outward spreading" is poor phrasing on our part, and this line has been removed from the manuscript, as has this entire section. We now discuss the growth of terrain near the ELA base level, and support this claim with our results from "progressive hypsometry" (Fig. 8).

Finally, the knickpoints that mark the break between glacial and fluvial landscapes around 3000 m are not the same knickpoints as those thought to represent the break between the uplifted relict landscape and steeper topography at ~2200 m.

*One could envisage the authors' model in case of continuous rapid uplift and fluvial downcutting, which is the case in Taiwan (I do not know the Costa Rica case sufficiently well to comment on this). But in this case, the scarp retreat would be independent of the glacial "buzz-cutting" and would happen anyway (which it does; pretty much every hill slope in the Taiwan Central Range is affected by landsliding). In that case, the glacially affected high-elevation low-relief parts of the landscape are just transients that are rapidly erased and one can question their significance for overall long-term landscape development. This part of the model clearly requires some more elaboration.*

We have addressed this comment thoroughly at various points throughout the revised manuscript. For example, we are now more explicit about our choosing to focus on the Talamanca Range and Central Range of Taiwan because glacial erosion there has left a very marginal imprint, particularly compared to mountain ranges such as the Merauke Range and the Mérida Range.

In the context of ten tropical mountain ranges, we call attention to the fact that both the Talamanca Range and the Central Range of Taiwan (as well as the Finisterre Range, Owen Stanley Range, and Crocker Range) are all capped by the cold-phase, despite being only marginally glaciated. We pose three possibilities for this coincidence, which are discussed on p. 22, Sec. 7.2, and illustrated in Fig. 2.

The scenario described in the comment above (*the glacially affected high-elevation low-relief parts of the landscape are just transients that are rapidly erased and one can question their significance for overall long-term landscape development*) is actually closest to the scenario we propose to be the most likely. The crux of how "significant" a role glacial erosion has played in the long-term evolution of a mountain range such as Taiwan is a function of 1) how close it is to a fluvially-driven steady state elevation in the absence of glacial erosion, and 2) whether it is likely that it has been glaciated more than once. These possibilities are discussed in detail in Sec. 7.2.

*More specific comments, tied to page/line number:*

*p. 2 / l. 15 (and elsewhere): some of the wording in the manuscript ("we add a new spin to the story . . .") makes it sound like the objective here is to "push" a "nice story" instead of seeking truth, which is what science is (should be) about. This is probably not the authors' intention and the writing is simply a bit too colloquial in places, but you should really try to avoid such phrasing.*

Our objective is certainly not to push a nice story. We have reworded colloquial phrasing throughout the manuscript.

*p. 3/ l. 20-24. The authors should be aware of a recent re-analysis (Schildgen et al., in press) that has shown the Herman et al. (2013) results to be flawed by a "spatial correlation bias", in which spatial variations in exhumation rates are translated into temporal increases by their model. Therefore, the thermochronometric record can no longer be used as support for increased erosion rates during Quaternary glaciations. Also, note that the Shuster et al. (2011) study argued for rapid glacial-valley incision (i.e. analogous to what Valla et al. (2011) argued for in the Western Alps) and does not pertain to glacial "buzz-cutting".*

Thank you for pointing us to this reference. We have rewritten this section of the introduction, and now reference this paper (p. 7, l. 5-11).

*p. 5 / l. 24: "narrative" – see comment on p. 2/l. 15 above.*

*p. 6/ l. 1-7: this needs to be backed up by field photos and a geomorphic map.*

Field photos and a geomorphic map are now included in Fig. 9.

*p. 7 / l. 1-5: similarly, a map of the Nanhudashan area showing the occurrence of these glacial forms would be useful.*

This is now included in Fig. 10.

*p. 7 / l. 20: Shuster et al. (2011) focused on glacial valley incision, not on cirque retreat.*

We have removed this reference from the discussion of cirque retreat.

*p. 8 / l. 27: "our conclusions are not affected by the choice of production rate or scaling"; without any justification, this is a rather empty statement. I would suggest to either delete it or to provide supporting data.*

The supporting data are included in the supplementary file, and simply show that different scaling regimes alter calculated exposure ages by <2 kyr. Since our central conclusion is that glacial valleys at Chirripó were subject to LGM erosion, we conclude this relatively small range of

variability does not affect our overall conclusion. We ultimately removed this line from the revised manuscript.

*p. 9 / l. 10-12: a slope map would help to demonstrate and justify the location of these erosional scarps.*

We did not find room for a stand-alone slope map, but we do give a sense of the presence of these scarps with the binary slope map superposed on the DEM in Fig. 9C and Fig. 10C as well as with field photos in Fig. 6 and Fig. 10.

*p. 10 / l. 18-20: can you elaborate on what this statement is based on?*

Elaboration is now provided on p. 17, l. 5-10.

We sought a way to quantify the effect of scarp encroachment into glacial catchments, specifically, what segment of glaciated valleys have been affected by scarp encroachment. To do so, we required some reference point for individual catchment outlet elevations, which we chose to be 3000 m. We chose this elevation on the basis that the lowest moraines observed extend to about 3000 m elevation.

*p. 11 / l. 12: "unrealistically" appears as a strange word choice for assessing data. What you probably mean is that this age, which is significantly younger than the LGM, implies that the surface must have been buried. Nothing unrealistic about that . . .*

We have reworded this sentence (p. 19, l. 4-6).

*p. 12 / l. 3: the glacial ELA elevation in Taiwan has not been demonstrated or even discussed at this point.*

Discussed above.

*p. 12 / l. 25-26: this statement requires justification.*

This line has been removed in the revised manuscript, but we have now documented more clearly that the highest landscapes in both Taiwan and Costa Rica are close to the cpELA, and upon careful examination of several such landscapes, we see clear evidence of glacial erosion.

*p. 14 / l. 6: what do you mean by "tile-scale"?*

1°x1° SRTM DEM tiles. Egholm et al. (2009) used these tiles in their global analysis.

*p. 14 / l. 10-12: I don't think this statement has been demonstrated. One could just as easily argue, even within the context of this model, that the mountain belt elevation hovers around an elevation that is set by the relative efficiency of tectonic uplift versus (glacial or fluvial) erosion – it is*

*lowered a bit during glacial times and uplifted during the transient post-glacial period of scarp encroachment.*

See comment about three landscape evolution scenarios above.

*p. 14 / l. 24-25: this is a fairly bold statement that extrapolates the findings and interpretations from Nanhudashan to all of the Taiwan Central Range. To do this, you would at a minimum need to show that the rest of the Central Range is equally affected by glacial erosion of the highest peaks and shows similar morphometry. In my understanding, glacial features in Taiwan have only been described from Nanhudashan.*

This sentence has been removed by a similar case is made in Sec. 7.2.

First, to clarify, LGM glacial features in Taiwan are best preserved at Nanhudashan, and for this reason we focused our analysis there. More ambiguous LGM glacial features have also been reported at Hsuehshan (Cui et al., 2002) and Yushan (e.g., Hebenstreit et al., 2011). We now discuss this on p. 12, l. 1-17.

We have proposed that an explanation for the relatively constant elevation of peaks throughout Taiwan—many of them close to the ELA—can be explained by a glacial erosion acting at different, isolated peaks throughout the Pleistocene. Wherever glacial erosion does occur, a low-sloping, transient glacial landscape is left behind and eventually wiped from the landscape by fluvially-driven scarp propagation.

We respectfully disagree with the statement that "at a minimum" we would need to show that "the rest of the Central Range is equally affected by glacial erosion." In our model context, glacial erosion stops the highest parts of the landscape at the ELA, and glacial landscapes are then removed by the flanking fluvial network by scarp encroachment, reducing their preservation potential. To infer glacio-fluvial height limitation in Taiwan we rely on similar evidence from comparable mountain ranges (such as the Talamnaca Range, Finisterre Range, Owen Stanley Range, and Crocker Range). This statement is thus not premised on our observations from Taiwan alone.

*p. 14 / l. 31-32: how would the glacial "buzz-cutting" "prime" the landscape for rapid horizontal scarp encroachment? See general comment above.*

See answer above with regard to scarp encroachment.

*Fig. 1: it would be nice to have an uncluttered DEM image with an elevation scale (as well as a horizontal scale and indications of latitude and longitude). The glacial extent and the location of the scarps could be moved to the satellite image of fig. 1a (or better, could be part of a geomorphological map). The inset location map is close to unreadable.*

We have made updates to the Costa Rica Fig. 9 to improve readability.

**Reviewer #2**

*General comments:*
*Most previous studies of mountain range height and glacial erosion have used correlations between ELA and max topography/hypsometric maxima along climatic gradients caused by temperature or precipitation to infer that glacial erosion influences mountain range height. To me such spatial correlations provide a stronger argument than the two isolated cases presented here. We know from global compilations of topography and ELA that many exceptions to the overall trend exist for numerous reasons. I therefore encourage you to expand your study and collect data from more tropical ranges. Do any of the tropical ranges stand high above the ELA? Or do the two cases documented here indeed represent a general pattern? That two selected ranges have heights that match the estimated ELA can easily be a coincidence. Even worse: Were the ranges selected for this study because they happen to have heights that match the ELA? You need to show us more data to answer such questions and to support the general points made.*

To be clear, the decision to focus on the Talamanca Range and the Central Range did not begin with their ELA-height match. Rather, we were initially struck by the following: even though global scale observations (e.g., Egholm et al., 2009, Fig. 1C, adapted and presented in the revised manuscript as Fig. 1) of the ELA-height match include the tropics, glacial erosion has *not been proposed as a mechanism for limiting tropical mountain height*. We now make this point clearly in Sec. 1.2 (p. 3-4).

The core revisions address this comment. We begin with a review of the evolution of thought on glacial limitation (Sec. 2.2, p. 5-7), and emphasize that glacial limitation is not generally accepted as a viable mechanism in the tropics (Sec. 2.3, p. 7). To reassess this claim, we now present analysis of ten tropical mountain ranges. We originally considered all high tropical mountain ranges, but ultimately excluded some mountain ranges which we thought would reduce the clarity of the analysis (Sec. 3.1.1, p. 7). The list of tropical mountains now includes:

1) Leuser Range, Aceh, Indonesia

2) Central Range, Taiwan

3) Talamanca Range, Costa Rica

4) Crocker Range, Borneo

5) Finisterre Range, Papua New Guinea

6) Owen Stanley Range, Papua New Guinea

7) Merauke Range, Papua

8) Mérida Range, Venezuela

9) Sierra Nevada de Santa Marta, Colombia

10) Rwenzori, East Africa

We present hypsometric analysis of all of these mountain ranges (Fig. 8), the details of which we discuss in response to a separate comment. We ultimately choose to focus on the Talamanca Range in Costa Rica and the Central Range of Taiwan because glaciation there has been particularly marginal (Sec. 3.2, p. 9-10).

*Regarding the topographical analysis you compute the hypsometry for individual catchments (focused hypsometric analysis) instead of simply computing the hypsometry of a large area (the full range, or anything above a certain elevation). While this may open for more detailed insights, it also has disadvantages when it comes to hypsometric maxima, because a catchment defined by flow routing should always have a hypsometric maxima somewhere in between the max and min elevations in the catchment. Hypsometry of a catchment may therefore differ from the hypsometry of a mountain range, which can have a hypsometric max close to base-level. Your use of catchments at different scales only partly address this issue, and to me mountain range hypsometry is just a simpler metric to understand and use.*

We have engaged with this comment a great deal, and address it on several levels. First, we describe the ways that hypsometry has traditionally been used to assess glacial limitation (p. 5, l. 21 – p. 6, l. 15; p. 13 Secs. 5.2). We then employ traditional (mountain range-scale) hypsometric analysis on the ten targeted mountain ranges.

A central question is whether the absence of a hypsometric maximum at the mountain range-scale is indicative of the absence of (significant) glacial erosion. When the ELA is a relatively high elevation, fluvial catchments must be large (in elevation range) for glacial erosion to take place at all. Thus the absence of a hypsometric maximum at the ELA on the large scale does not indicate the absence of glacial erosion, or even glacial limitation (or glacio-fluvial limitation for that matter).

We thus are left with the following problem: what scale of analysis is appropriate to assess the role of glacial erosion in limiting mountain height? We argue that different scales of analysis are needed to assess the overall significance of glacial landscapes in environments like those found in the tropics.

We introduce a new method of hypsometric analysis that we call "progressive hypsometry" (PH) in Sec. 5.2.2, at which point we describe the algorithm in detail. We then implement PH in the ten selected tropical mountain belts. We pose this method as a solution to the problem of scale in hypsometric analysis. Rather than choosing one scale, either large or small, and checking for a hypsometric maximum at the ELA, we find the hypsometric maximum of catchments at virtually all scales in a targeted mountain range.

To summarize, progressive hypsometry can characterize the fine scale topographic patterns of entire mountain ranges, and can reveal features that go missed in tile hypsometry (Fig. 8a2-j2). In all of the mountain ranges we analyze, mountain range-scale hypsometry shows very little area near the ELA. Yet, nine of ten show evidence of glacial erosion at high elevations, and in some cases particularly strong glacial erosion. We discuss these results extensively in Sec. 7.1, p. 19-22.

*Alternatively, you could also compare with focused hypsometries of catchments where there are no signs of glacial erosion. Do they have the same type of maximum or are they notably different?*

Progressive hypsometry allows us to compare the hypsometry of virtually all catchments in each mountain range.

*It would be useful to also see longitudinal profiles of valleys with and without evidence of glacial erosion.*

We have looked at the longitudinal profiles of glaciated and unglaciated valleys in Costa Rica and Taiwan, but, in light of the new data provided in the revised manuscript, we consider hypsometry to be the most helpful to contextualize glacial influence (e.g., Fig. 8).

*I recommend that you also address the height of the ridges above the ELA. The ridges on the plateau are rather low and I would expect them to be higher, if glacial erosion around LGM was the main erosion mechanism at high elevation. Pedersen et al. (Geomorphology 122, p. 129-139, 2010) showed how ridge height above ELA seemingly depends on the rate of tectonic uplift. Tectonic uplift rates are high in both these ranges, so what keeps the ridges down to few hundred meters above the estimated ELA? Could it be periglacial slope processes, and would they have enough time to operate in the Holocene?*

Thank you for reminding us about this important reference, which is highly relevant to our work.

Unfortunately, this comment is difficult to address in a straightforward manner. As we describe in Sec. 7.1, p. 21, l. 3-23, we propose that the growth of tropical glacial landscapes is governed by the following three factors:

(i)     the volume and pattern of rock uplift through the cpELA;
(ii)    the efficacy of glacial erosion; and
(iii)   fluvially-driven destruction of glaciated terrain.

Pedersen et al. (2010) invoke a steady state balance between rock uplift and glacial erosion to explain the correlation between ridge height and uplift rate in glacially eroded landscapes at the mid-latitudes (p. 136). In the mid-latitudes, it is easier to envision how this relationship develops. Our revised manuscript highlights that tropical glacial landscapes evolve in a rather different way, so this particular model is not necessarily applicable to the mountains we analyze.

*More specific questions:*

*Page 3 Line 29: I do not see how it can be a provocative statement that glacial erosion limits the height of mountains – erosion does that. Please rephrase to explain the provocative part.*

Thank you for bringing this to our attention. We now specify on p. 6, l. 26-27 that it is not universally accepted that *ice-driven* erosion, specifically, limits mountain height. We expand on this idea on p. 6 l. 26- p. 7 l. 8 in the revised manuscript.

*Page 4 line 5-10: This paragraph unfortunately repeats a misunderstanding that I think started with Hall & Kleman (2014): The glacial buzzsaw mechanism does not rely on horizontal erosion, and I do not think that any of the computational landscape models that you cite (e.g. Anderson, 2006; Egholm et al., 2009; MacGregor et al., 2009) even have horizontal erosion. The link between ELA and hypsometry arises because (vertical) glacial erosion is downwards limited by the mass balance of the glaciers (Egholm et al., 2009). Small glaciers do not erode deeper than the ELA because they cannot exist there. Larger glaciers can, however, because the ice flux into them keeps them alive well below the ELA. That larger glaciers cut deeper and faster than cirque glaciers is therefore not surprising, and not at all in conflict with models for the glacial buzzsaw. These two elements of a glacial landscape go hand in hand.*

This paragraph was in reference to Valla et al. (2011) who claimed that evidence of rapid glacial incision below the ELA "contradicted" the concept of the glacial buzzsaw, but your comment demonstrates clearly that their findings are not necessarily such a contradiction. We have rewritten this section entirely.

We now focus on the concept of a glacial base-level, which Egholm et al., 2009 reference in a similar way to the comment above. This idea is developed in Sec. 1.1 and Sec. 2.2.

*Page 6, line 10: It would be good to have an uncertainty estimate for the ELA. It is important here because the differences in hypsometric maxima are rather small.*

We ultimately chose to restrict the ELA to a single benchmark value in this segment of the analysis because we were interested in whether variability in the modal elevation of glacial catchments could be explained by scarp erosion. These landscapes are not large enough for substantial variation in the ELA, and we propose that the variability we do see is driven by post-glacial erosion. Our findings support this claim.

*Page 7, line 21: Why not record the aggregate of many valleys? Sounds good to me.*

Hypsometric maxima at the "aggregate" scale are effectively recorded in progressive hypsometry.

*Page 9, line 31: This is where the uncertainty on the ELA becomes relevant.*

See comment above.

*Page 11, line 4: I do not think that you are constraining the timing of glacial erosion here. Your (few) boulder samples may constrain timing of deglaciation, but the (even fewer) bedrock samples do not show any clear pattern.*

This line has been rephrased.

*Page 14, line 30 and many other places including the title: Why not just write "erosion" instead of "buzzcutting"? I don't think we really need more "buzzwords" than we already have.*

We have removed all mention of "buzzcutting" and stick with "glacial limitation". This terminology is presently used in the literature, such as in Egholm et al. (2009) "Glacial effects limiting mountain height".